# Foveal neurons of the monkey superior colliculus signal trans-saccadic prediction errors

Tong Zhang[1,2☯], Amarender R. Bogadhi[1,2,3☯], Ziad M. Hafed [1,2*]

**1** Werner Reichardt Centre for Integrative Neuroscience, University of Tübingen, Tübingen, Germany, **2** Hertie Institute for Clinical Brain Research, University of Tübingen, Tübingen, Germany, **3** Neuroscience and Mental Health Discovery Research, Boehringer Ingelheim Pharma GmbH & Co. KG, Biberach, Germany

☯ These authors contributed equally to this work.
* ziad.m.hafed@cin.uni-tuebingen.de

## Abstract

Across saccades, neurons in retinotopically organized visual representations experience drastically different images, but visual percepts remain stable. Here we investigated whether such stability can be mediated, in part, via prediction-error signaling by neurons processing post-saccadic visual images. We specifically recorded from foveal superior colliculus (SC) neurons when a visual image only overlapped with their response fields (RF's) after foveating saccades but not pre-saccadically. When we rapidly changed the target features intra-saccadically, the foveal neurons' post-saccadic visual reafferent responses were elevated, even though the neurons did not directly sample the pre-saccadic extrafoveal target features. This effect did not occur in the absence of saccades, and it also scaled with the extent of the introduced intra-saccadic image feature discrepancies. These results suggest that foveal SC neurons may signal a trans-saccadic prediction error when the foveated image stimulating them is inconsistent with that expected from pre-saccadic extrafoveal representations, a potential perceptual stability mechanism.

## Introduction

Sensory systems benefit from spatial and temporal continuity of incoming information streams. Often, such continuity is disrupted, whether through exogenous environmental factors or through self-movement. One example such disruption occurs every time we generate a rapid eye movement. Across saccades, which we typically use to foveate relevant extrafoveal image patches, retinotopic representations of foveal visual inputs can experience drastically different images. For example, when viewing a painting of a natural scene, we might generate a saccade from one foveated image patch, say containing a flower, to another one, say containing a leaf. In this case, the pre-saccadic extrafoveal image of the leaf is rapidly translated to become the post-saccadic foveal image; thus, the pre- and post-saccadic foveal visual images

**Data availability statement:** All relevant data are within the paper and its Supporting Information files.

**Funding:** We were funded by the Deutsche Forschungsgemeinschaft (DFG; German Research Foundation; www.dfg.de) through the Special Priority Programme: "SPP 2411 Sensing LOOPS: cortico-subcortical interactions for adaptive sensing" (project numbers: 520617944 for Z.M.H., 520283985 for Z.M.H. and T.Z., and HA 6749/11-1 for Z.M.H. and T.Z.). We were also funded by the DFG project BO5681/1-1 (for A.R.B. and Z.M.H.). The funders did not play any role in the study design, data collection and analysis, decision to publish, or preparation of the manuscript.

**Competing interests:** The authors have declared that no competing interests exist.

**Abbreviations :** FEF, frontal eye field; LIP, lateral intraparietal area; RF, response field; ROC, receiver operating characteristic; SC, superior colliculus.

are decidedly different from each other. Despite that, our percept of the portrait remains seamless.

The question of perceptual stability across saccades has intrigued scientists for many decades [1–8]. Among the theories invoked for such stability are ones utilizing a form of prediction (meant in the phenomenological sense) of the visual consequences of eye movements [3,9]. Conceptually, such prediction may be thought of as a template matching between newly incoming foveal visual information and the expected appearance, obtained a priori, of the post-saccadic foveal image [3,9]. Such matching could arise, for example, by relaying a pre-saccadic extrafoveal preview of the eye movement's target [10] to neurons processing the newly foveated visual evidence. Here, we searched for a neural correlate of such a process in the visual responses of foveal superior colliculus (SC) neurons. We specifically asked whether these neurons' post-saccadic reafferent responses (after target foveation) are sensitive to the pre-saccadic extrafoveal image appearance, even when such extrafoveal visual input never activated the neurons' classic response fields (RF's) pre-saccadically. If so, then this would suggest a potential trans-saccadic dispatching of extrafoveal pre-saccadic visual information to the foveal SC.

The SC is particularly attractive for exploring hallmarks of extrafoveal-to-foveal trans-saccadic integration. First, SC neurons exhibit sufficiently varied feature tuning properties [11], allowing us to investigate reafferent visual responses for different image manipulations. Importantly, such feature tuning endows extrafoveal SC neurons with integrated evidence about the extrafoveal appearance of saccade targets [11]; while likely not enough to process very fine details [11–13], this integrated evidence can provide a sufficient [13–15] pre-saccadic preview. Second, the SC possesses a putative mechanism for relaying exactly such an extrafoveal preview to the rest of the visual system (including foveal representations) at the time of saccades [16–19]. In particular, SC motor bursts, time-locked to saccade onset [20], embed within them a representation of the visual properties of the extrafoveal saccade target [16,17,19]. Thus, whether through intra- or extra-collicular connections, SC foveal representations [21] can be "informed" about extrafoveal saccade target appearance, despite not being directly visually-stimulated by the saccade target pre-saccadically. Finally, the SC signals salience [22–25]; hence, trans-saccadic integration by foveal SC neurons would be functionally useful to signal failures of image stability. In other words, potential inconsistencies between experienced versus expected foveal images constitute salient events worthy of flagging, so detecting them is desirable.

Here, we show that foveal SC neurons are indeed sensitive to the pre-saccadic extrafoveal target appearance, despite never being stimulated by the extrafoveal targets before the eye movements. We also show that such sensitivity is contingent upon saccade generation, depends on extrafoveal target visibility, and scales with the extent of sensory conflict experienced between pre-saccadic extrafoveal and post-saccadic foveal image appearances. We suggest that the SC is endowed with mechanisms for signaling salient events in the temporal domain [26–28], such as happens trans-saccadically, and not just in the image-based one [22–25].

## Results

### Foveal superior colliculus neurons detect unexpected trans-saccadic stimulus changes

We recorded from foveal SC neurons while monkeys generated visually-guided saccades to extrafoveal images containing different spatial frequency textures [12,13,16,29] ("Methods"). In our basic condition, the saccade target was a circular patch containing one of two possible spatial frequencies, and it appeared either to the right or left of fixation ("Methods"). A saccade "go" signal was provided a few hundred milliseconds later, after which the monkeys foveated the circular patch (Fig 1a). The patch itself never moved on the display, but it was rapidly translated on the retina by saccades, eventually entering the RF's of our recorded foveal SC neurons and visually stimulating them.

We employed a strict selection of neurons to include in our analyses ("Methods"). Specifically, we ensured that the neurons were not visually-stimulated by the saccade target except after eye movement generation. Thus, we only accepted neurons with foveal RF's that were not visually-driven (before the saccades) by the extrafoveal image patch (RF's were estimated through a mapping task run before our main paradigm; "Methods"). Fig 1b shows an example such RF, with the white circle indicating the size of the saccade-target image patch (were the eye to be perfectly aligned with the patch's center). In the actual experiment, this patch was centered at an eccentricity of 8 deg before the saccade, and was thus far from the neuron's RF; however, the same patch was large enough to clearly visually drive the neuron after the eye

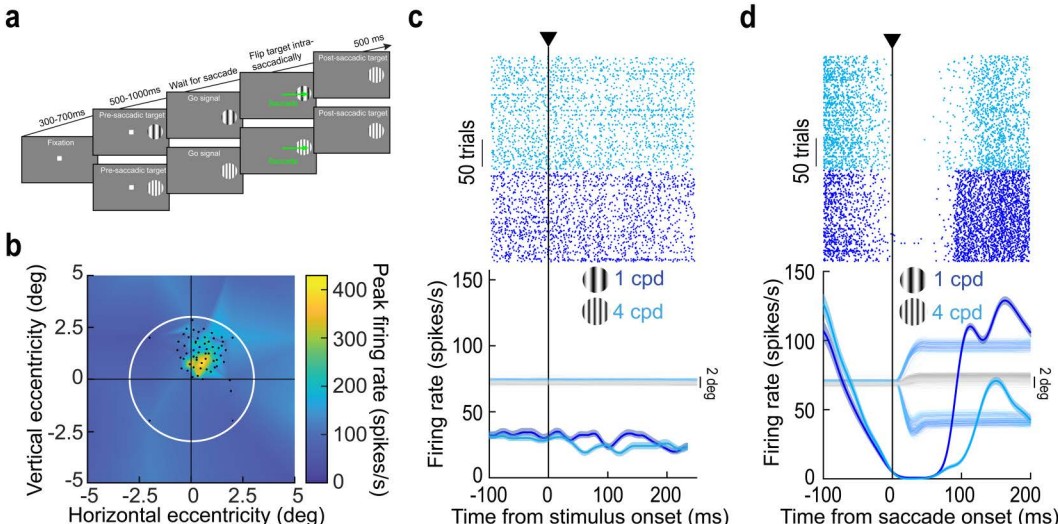

**Fig 1. Testing for trans-saccadic prediction error signaling by foveal SC neurons.** (a) Monkeys generated a delayed, visually-guided saccade towards an extrafoveal target. We used a delayed paradigm to make sure that there was a stable visual image upon saccade generation. In some trials, the saccade target was unchanged throughout the whole trial (high spatial frequency grating embedded within a circular patch for this shown example). In other trials, we detected saccade onset and immediately flipped the saccade target to another feature (from a low to a high spatial frequency texture in the shown example; "Methods"). (b) We only selected foveal SC neurons with response fields (RF's) not extending towards the pre-saccadic extrafoveal stimulus location. In this example, the RF was almost entirely contained within <2 deg eccentricity. Each black dot is a stimulus onset location during RF mapping, and the white circle (3 deg radius) shows the extent of the saccade target if it was perfectly foveated post-saccadically. The target covered the RF post-saccadically but not pre-saccadically. The z-axis indicates the visual response strength of the neuron at each stimulus location ("Methods"). (c) Activity of the same example neuron when the extrafoveal stimulus appeared. There was no visual response, confirming that the pre-saccadic stimulus did not drive the neuron. The faint lines show eye position traces (the monkey was still fixating the central fixation spot). (d) After saccades, the neuron exhibited a visual reafferent response to the image feature entering its RF. Note how the neuron completely paused intra-saccadically and only responded after the new visual evidence entered its RF (there was additionally a pre-saccadic transient elevation before the pause due to the release of fixation, which we do not focus on in this study). Note also that the neuron was visually-tuned to the low spatial frequency [12,29]. The faint lines indicate eye position traces, showing how the neural modulations were time-locked to saccade onset. Error bars denote SEM, and numbers of trials are evident from the rasters. The underlying data for **b–d** are included in S1 Data.

movement has ended (i.e., after the patch was retinally translated towards the fovea by the saccade). We confirmed these two points by measuring the neuron's activity after extrafoveal stimulus onset and also upon saccade generation. In the former case, there were no visually-evoked bursts (Fig 1c shows the neuron's activity when extrafoveal image patches containing two different spatial frequency textures appeared; "Methods"). If anything, there was a transient reduction in baseline activity, particularly for the high spatial frequency texture. Such transient reduction has been observed before upon extrafoveal stimulus onset (for examples, see ref. [30]), and it can aid in subsequent winner-take-all processes needed to recruit the extrafoveal saccade-related motor bursts [31]. Upon saccade generation, the neuron now exhibited a visual reafferent response caused by translating the saccade-target image into its RF (Fig 1d). Importantly, by employing two different spatial frequencies, we ensured that this response was visual in nature, because it depended on the visual appearance of the foveated saccade target [29]. For example, the reafferent response of the neuron of Fig 1b and 1c was earlier and stronger for the low spatial frequency texture, as might be expected [29] (Fig 1d).

Having established that our neurons were not visually stimulated by the pre-saccadic extrafoveal stimuli (S1 Fig), we next analyzed the situation in which there was an image mismatch between the experienced post-saccadic visual stimulus and the pre-saccadic (extrafoveal) target image. For each saccade to a given image feature (low or high spatial frequency), we had two conditions: no change in the image feature across the saccade (control), or an instantaneous intra-saccadic change (to the other spatial frequency; "Methods"). For example, if the pre-saccadic image had a high spatial frequency texture, then, during the saccade, we flipped it to the low spatial frequency texture. Thus, the foveal neurons experienced a visual feature different from that expected from the extrafoveal pre-saccadic feature, but these neurons were never directly stimulated by the extrafoveal feature pre-saccadically. We used instantaneous eye position to detect saccade onset and updated the display accordingly (either fictively in the control condition or actually; "Methods"). This allowed us to compare the visual reafferent response for the same image feature in the fovea (e.g., landing on a low spatial frequency), but either when the extrafoveal pre-saccadic image was the same (low spatial frequency) or different (high spatial frequency).

Fig 2a shows example radial eye speed traces for rightward saccades from the same session of the example neuron of Fig 1. Here, we aligned all traces to the end of the stimulus update on trials in which the saccade target flipped from containing a high to a low spatial frequency texture during the saccade (similar results were obtained for the other image flip condition). As can be seen, the stimulus update was intra-saccadic, and the eye speed by the end of the stimulus change was systematically higher than approximately 100 deg/s. Thus, given that the gratings were vertically oriented and the saccades were predominantly horizontal, this ensured substantial blurring of both spatial frequencies on the retina by the ongoing eye movements (and during the image change). In Fig 2b, we plotted the visual reafferent response of the same example neuron of Fig 1, but now aligned to the time of intra-saccadic stimulus update completion in all trials (note that control trials had fictive updates, allowing for similar temporal alignment in the two conditions). In both shown conditions, we were measuring the neuron's visual reafferent response for the same post-saccadic foveal stimulus (low spatial frequency); the only difference is that, in one case (blue), the extrafoveal pre-saccadic image was the same (control; low-to-low spatial frequency from pre-saccadic to post-saccadic image appearances); in the other (yellow), it was different (high-to-low spatial frequency change). The neuron's visual reafferent response was elevated when there was a mismatch in the pre-saccadic image, even though the foveated visual feature was the same as in the control condition. This was the case even though the extrafoveal image location was outside of the neuron's RF before the saccade (Fig 1).

The same observation was also made when the post-saccadic foveal visual image feature was the high spatial frequency (Fig 2c). Again, the neuron's reafferent response was elevated (relative to control) when there was a mismatch with the pre-saccadic extrafoveal saccade-target appearance (i.e., when the pre-saccadic image was a low spatial frequency). Note that there was no other change in the firing patterns of the neuron in Fig 2b and 2c, for example in the reafferent response timing on stimulus change trials. This suggests that we were not observing a simple superposition of the original reafferent response and an additional sensory drive by the stimulus flip event (or other transients associated with

   

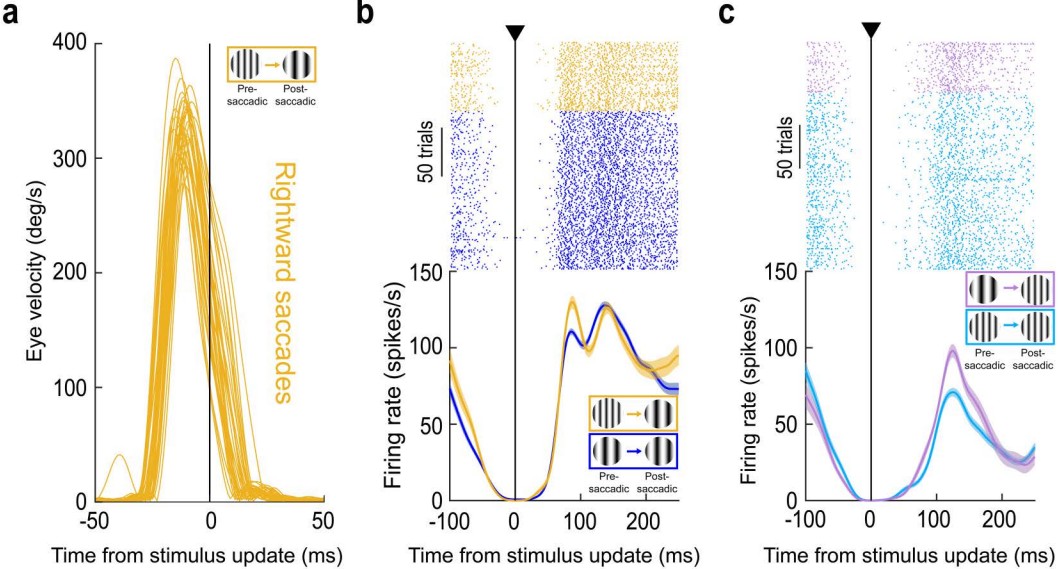

**Fig 2. Signaling of trans-saccadic prediction error by the foveal SC.** (a) Radial eye speed traces from the same session as in Fig 1b–1d when the monkey was generating foveating saccades. Here, we aligned the traces to the end of the stimulus update event (on trials in which the pre-saccadic extrafoveal texture was of high spatial frequency and the post-saccadic foveal feature was a low spatial frequency texture). Eye speed was consistently >100 deg/s during stimulus updates (also see S2a and S2b Fig demonstrating that we always achieved stimulus updates intra-saccadically and with large eye speeds, maximizing stimulus blurring). (b) Relative to control (blue; same data as in Fig 1), the initial reafferent response of the neuron was elevated (yellow). This happened even though the neuron was not stimulated by the pre-saccadic extrafoveal stimulus (Fig 1). (c) Similar observations when the foveated image feature contained a high spatial frequency; in this case, the control condition also contained a high spatial frequency pre-saccadically. Thus, the neuron's visual reafferent response was systematically elevated when the experienced post-saccadic visual feature was different from that expected based on the pre-saccadic extrafoveal target appearance; this was despite the lack of visual stimulation of the neuron pre-saccadically by the extrafoveal saccade target (Fig 1). Note also how the reafferent response timing was similar on control and stimulus-change trials, confirming that our temporal alignment (to the stimulus update time) was similar across the different conditions; thus, the different reafferent response magnitudes across conditions were not an artifact of different temporal jitters across them. Error bars denote SEM, and numbers of trials in each condition are evident from the individual traces and rasters. The figure's underlying data are included in S2 Data.

saccade execution); rather, there was a modulation of the otherwise-normal reafferent response, as if it was more salient than expected. Thus, this example foveal SC neuron signaled a trans-saccadic change in the saccade-target appearance, even though it did not "see" the saccade target pre-saccadically.

Across neurons and sessions, we confirmed that all of our stimulus updates were intra-saccadic and with high eye speeds maximizing the likelihood of large retinal image blur (S2a and S2b Fig). We also confirmed that saccade speeds (including post-saccadic drift [32]) (S2c and S2d Fig) and radial amplitudes (S2e and S2f Fig) were not altered by the intra-saccadic stimulus changes. And, we confirmed that the neurons' RF hotspot locations were foveal (S3 Fig).

At the level of reafferent responses, there was a systematic elevation of their magnitudes, for each spatial frequency, if the pre-saccadic image contained a different feature from that experienced foveally after the saccade. This result was assessed in multiple ways. First, we calculated the normalized population firing rate in each condition, and visualized the different conditions in Fig 3a and 3b ("Methods"). As can be seen, the population dynamics behaved very similarly to those of the example neuron of Fig 2b and 2c. Second, we performed receiver operating characteristic (ROC) analysis, like we recently performed [16]. Elevations in the area under the ROC curve indicated elevated firing rates in intra-saccadic stimulus change trials. As can be seen, whether the post-saccadic image was a low (Fig 3c) or high (Fig 3d) spatial frequency, there was always an elevation of reafferent visual response amplitudes on stimulus change trials relative to pre-saccadically (error bars denote 95% confidence intervals). We then plotted individual neuron reafferent responses

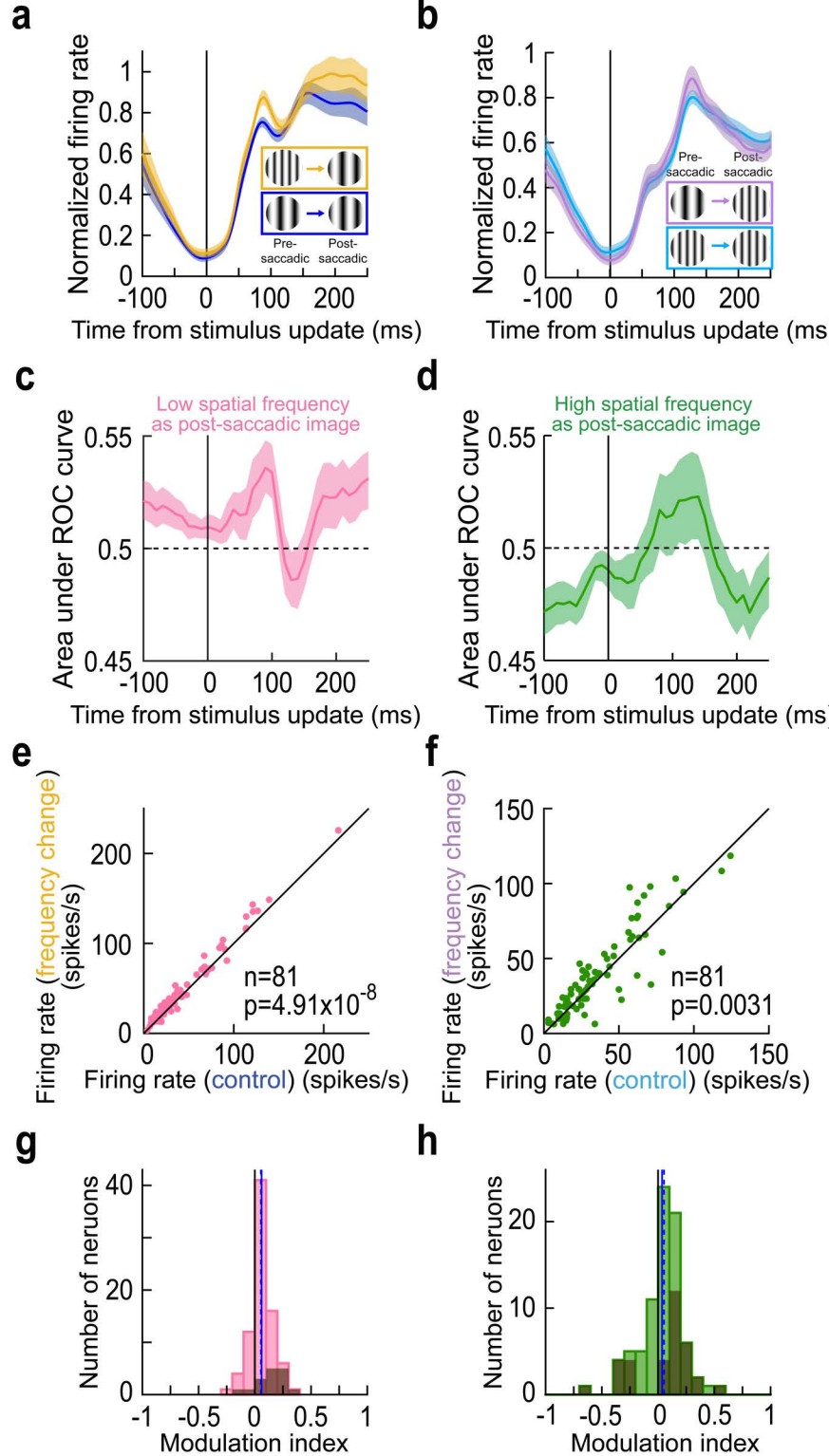

**Fig 3. Consistency of trans-saccadic error signaling by foveal SC neurons.** (a, b) Similar to Fig 2b and 2c but now showing the normalized firing rate of the population. Each neuron's results were normalized before averaging across neurons ("Methods") and error bars denote SEM across neurons. The example neuron results of Fig 2b and 2c were representative of the whole population. (c, d) ROC analyses ("Methods"; also see ref. [16]) comparing

intra-saccadic change trials to control trials across the population. Elevations in the area under the ROC value indicate higher firing rates on intra-saccadic change trials. For both low and high spatial frequency foveated targets, there was an elevation in area under the ROC curve post-saccadically when compared to pre-saccadically (and also surpassing the 0.5 chance level). Note that the values below 0.5 at negative times in d reflect an earlier transient associated with the release of fixation, which is not the focus here. Error bars denote 95% confidence intervals. (e) Reafferent response strength ("Methods") on trials with an intra-saccadic spatial frequency change vs. reafferent response strength on control trials. In both cases, the foveated image feature was the low spatial frequency texture within the circular patch of the saccade target. There was a systematic elevation of reafferent response strengths across the population when there was a visual feature inconsistency across saccades. (f) Similar results for the case in which the foveated texture had a high spatial frequency. (g, h) Neural modulation indices ("Methods") for the results in e, f. For both foveated image features, there was a systematic elevation of foveal neuron visual reafferent response strength when the experienced post-saccadic image was inconsistent with that expected from the pre-saccadic target appearance. In each panel, the solid vertical line indicates the mean of the distribution, and the dashed vertical line indicates the median. Dark histograms indicate the neurons that were individually significant within a session ("Methods"). The figure's underlying data are included in S3 Data.

as scatter plots (Fig 3e and 3f). For each neuron, we measured the strength of the visual reafferent response (peak firing rate in a prescribed interval after the end of the real or fictive stimulus update; "Methods"). We did this twice: once when the pre-saccadic extrafoveal visual image was the same as that stimulating the foveal RF after the saccade (control), and once when the pre-saccadic extrafoveal image was of a different spatial frequency (intra-saccadic image change). We then plotted the two values against each other. Fig 3e shows the condition when the foveated stimulus had a low spatial frequency texture; the reafferent response to this texture was elevated if it was inconsistent with the extrafoveal pre-saccadic texture's spatial frequency ($p = 4.91 \times 10^{-8}$; Wilcoxon signed-rank test). Fig 3f shows the results from the same neurons when the foveated stimulus was a high spatial frequency and the extrafoveal stimulus in the image flip condition was a low spatial frequency ($p = 0.0031$; Wilcoxon signed-rank test). In both cases, the reafferent response expected from a particular visual feature in the RF (low or high spatial frequency) was systematically elevated when the visual feature was unexpected based on the extrafoveal pre-saccadic feature. This is a hallmark of prediction-error signaling.

Finally, we also created a neural modulation index for each foveated image spatial frequency (Fig 3g and 3h; the dark histograms indicate the data for the neurons that were individually significant within a session; "Methods"). This index was calculated by subtracting the reafferent response strength on the control condition from that in the image flip condition and then dividing by the sum of response strengths ("Methods"). Again, there was a systematic elevation of reafferent response strengths when the experienced post-saccadic foveal visual image feature was not consistent with that expected from the pre-saccadic extrafoveal visual appearance of the saccade target. Statistically, comparing population modulation indices to zero revealed significant elevations for both the low ($p = 1.47 \times 10^{-7}$; Wilcoxon signed-rank test) and high ($p = 0.0119$; Wilcoxon signed-rank test) spatial frequency foveated targets. Thus, foveal SC neurons signaled the occurrence of intra-saccadic stimulus changes, a kind of prediction error from the expected pre-saccadic extrafoveal target image appearance.

We also considered the possibility of systematic differences in landing positions across pre-saccadic target spatial frequencies and sessions. These did indeed exist (due to expected variances in saccades), but the retinotopic positions of the foveated image patches were always encompassing our neurons' RF hotspot locations (S3a Fig). Moreover, variations in landing positions by the saccades across conditions (thus causing variations in foveated retinotopic image positions relative to the foveal RF's) would not be expected to always systematically elevate reafferent visual responses; rather, they might equally likely also diminish responses if the eye landing position resulted in a sub-optimal visual stimulation of the RF by the image patch. This was clearly not the case in our data (Fig 3).

Therefore, our results so far indicate that, even though our foveal SC neurons were not directly visually stimulated pre-saccadically by the extrafoveal saccade targets, they were still, nonetheless, sensitive to these targets' pre-saccadic appearances: intra-saccadically introduced conflicts in post-saccadic target appearances resulted in elevated reafferent response magnitudes. Such elevation, under prediction error situations, is a hallmark of predictive coding in a variety of brain areas [7,8,33–36].

## Independence of trans-saccadic error signaling from neuron preference

We next checked whether an individual neuron's preference for a particular foveated image feature dictated whether the neuron was sensitive to a conflict with pre-saccadic image features. We assessed neuron preference from the control conditions of Figs 1–3: we measured the reafferent response strength when foveating a low spatial frequency (that was unchanged across the saccade) and compared it to the case when the saccade landed on a high spatial frequency (again with no intra-saccadic change). In our database, 30/81 neurons (37.04%) had numerically stronger visual reafferent responses when the foveated image feature was a high spatial frequency, whereas 51/81 neurons (62.96%) responded more strongly for the low spatial frequency patch. This is consistent with SC neurons generally preferring low spatial frequencies [12,29] (Fig 4a). Next, we plotted the results of Fig 3e and 3f against each other. Specifically, we measured, for each neuron, the difference in reafferent response strength between the low-to-high image flip condition and the corresponding control scenario (high-to-high), and we plotted it against the difference between the high-to-low image flip condition and the corresponding control condition (Fig 4b). There was no correlation between the two measures ($r = 0.0062$; $p = 0.956$), suggesting that we were not merely observing a single, universal sensitivity (per neuron) for just any intra-saccadic image update. Moreover, when we colored the neurons according to whether they preferred high or low spatial frequencies in Fig 4a, we found that neurons from both groups could signal a trans-saccadic inconsistency in foveated image appearance (by elevated reafferent response strengths; Fig 4b), and for both types of spatial frequency changes. For example, 38/51 neurons (74.5%) preferring the low spatial frequency had an elevated reafferent response when the pre-saccadic image was of a high spatial frequency. For the neurons preferring the high spatial frequency, 17/30 neurons (56.7%) signaled a trans-saccadic inconsistency by elevating their reafferent response strengths (even though this proportion was less than for neurons preferring the low spatial frequency, the trend was still in the same direction).

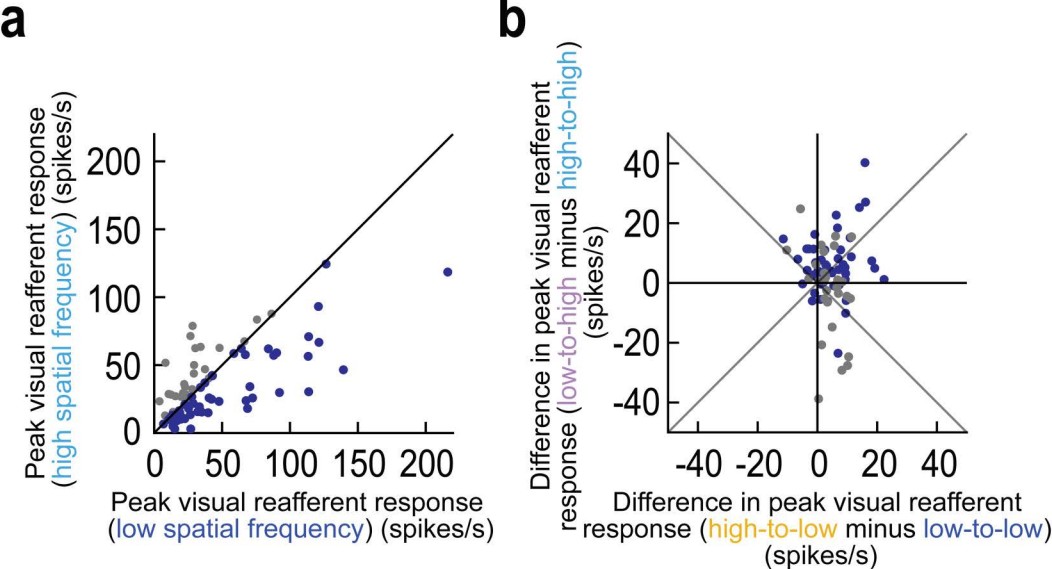

**Fig 4. Elevations of foveal reafferent response strengths with trans-saccadic image inconsistencies for both neurons preferring low and neurons preferring high spatial frequency textures.** (a) We assessed neuron preferences for low or high spatial frequency textures based on their reafferent responses in the control trials. More neurons preferred low spatial frequencies, as might be expected [12,29]. The different colors are used to help in classifying the different neuron groups in b. (b) For the same neurons, we plotted whether they elevated or decreased their reafferent responses on image-flip trials. Neurons with both preferences in a could elevate their reafferent responses with trans-saccadic image inconsistencies. Thus, there was no systematic dependence of the results of Figs 2–3 on neuron preferences. The figure's underlying data are included in S4 Data.

Thus, the results of Figs 2 and 3 were a general property of our population, and could happen regardless of individual neuron preferences for the foveated image features.

In Fig 4b, there were additional interesting differences in the foveal neurons' abilities to signal intra-saccadic stimulus changes in their reafferent responses. For example, when landing on a low spatial frequency image patch (x-axis), trans-saccadic error signaling seemed to be slightly more systematic than when landing on a high spatial frequency image patch, independent of neuron preferences (79.01% of all neurons elevated their reafferent response strength in the former case, as opposed to 67.9% in the latter, but this difference was not significant; $p > 0.5$; binomial distribution fit). In addition, going from a low spatial frequency pre-saccadic extrafoveal image to a high spatial frequency post-saccadic foveal image had neurons with more marked modulations in either direction (elevations or reductions relative to control; y-axis). These observations might suggest two possibilities. One, it could be that the SC might favor certain types of trans-saccadic image inconsistencies to "flag" by its foveal neurons (such as high-to-low spatial frequency image flips). Alternatively, since high spatial frequencies are significantly less visible and less salient than low spatial frequencies extrafoveally (due to visual acuity and contrast sensitivity functions of the visual system), a flip from high to low spatial frequency may elicit a more robust prediction error signal (on average) than a flip from an already highly salient extrafoveal image feature (low spatial frequency) to a less salient one. In a later experiment described below, we show evidence consistent with this latter possibility.

## Trans-saccadic error signaling does not occur with simulated saccades

To check whether a saccade was needed for foveal neurons to signal a trans-saccadic inconsistency between extrafoveal and foveal visual image appearances, we also ran a simulated saccade condition. In this condition, the monkeys always maintained gaze fixation, and we rapidly translated the extrafoveal image patch (initially appearing to the right or left of fixation at the same eccentricity as in the active saccade task) towards the fovea ("Methods"). On some trials, the patch had a fictive image change (control), and, on others, it had a real one, just like in the saccade task. Naturally, due to our discretized (both spatially and temporally) display, we could not mimic the exact saccadic eye movement profile, but we used a similar duration and peak speed of image translations ("Methods").

During maintained fixation, externally translating extrafoveal images into the neurons' foveal RF's did not elevate the neurons' visual responses to the stimuli on the image flip conditions relative to control; that is, there was no added spiking when the experienced foveal image feature was different from the pre-translation extrafoveal one. Consider, for example, Fig 5a, 5b, showing the responses of the same neuron of Figs 1 and 2 but in the simulated saccade condition. First, it should be noted that the neuron's activity was not completely paused during the image translation phase of the trials (compare Fig 5a and 5b to Fig 2b and 2c around the time of the stimulus update event). This is expected: the image translation in this condition was no different (conceptually) from a stimulus onset event used in many visual neurophysiology experiments; thus, the neuron just maintained its baseline firing rate until new sensory evidence entered into its RF. On the other hand, during real saccades, extrafoveal neurons were bursting and, simultaneously, foveal neurons paused [30,37–39]. More importantly, the same neuron's response to the afferent sensory input itself (the circular patch entering its RF) was the same, for a given spatial frequency, irrespective of the image flip condition. That is, when the stimulus entering the RF was a low spatial frequency, it did not matter whether the initial extrafoveal stimulus was a low or high spatial frequency (Fig 5a). Similarly, when the stimulus entering the RF was a high spatial frequency, it did not matter whether the extrafoveal stimulus was the same or different (Fig 5b). We also assessed this statistically: we measured the peak firing rate in a prescribed interval ("Methods") from (fictive or real) stimulus update time; whether the final foveal stimulus was of low or high spatial frequency, there was no difference in visual response strength relative to the corresponding control condition ($p = 0.1303$; Wilcoxon rank sum test comparing low-to-low and high-to-low; and $p = 0.1260$; Wilcoxon rank sum test comparing high-to-high and low-to-high trials). Thus, besides confirming our earlier assertion that our foveal neurons were not sensitive to extrafoveal image appearance during fixation, these results indicate that

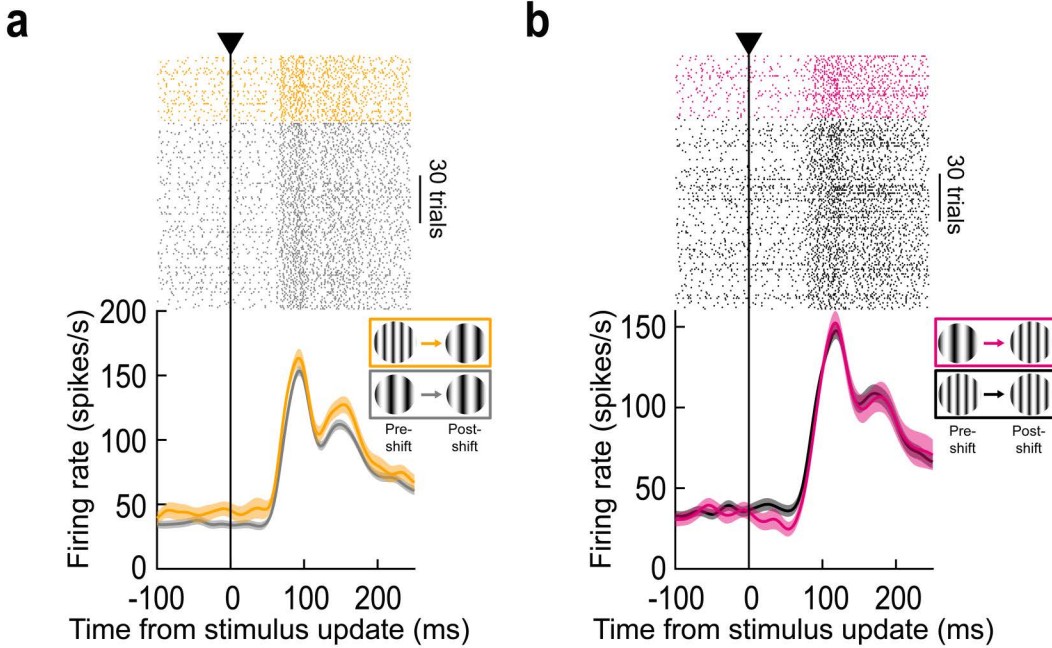

**Fig 5. Lack of foveal neural response elevations in the absence of real saccadic eye movements.** (a, b) Responses of the same example neuron of Figs 1 and 2, but this time in the absence of saccades. The monkey maintained gaze fixation, and we rapidly translated the extrafoveal image patch to the fovea ("Methods"). This resulted in a visual response to the patch entering the neuron's RF, and there was no prior pause in neural activity because there was no actual saccade generation (compare to Figs 1 and 2). Importantly, whether there was an intra-translation image flip or not, the neuron simply reflected the visual appearance of the final foveal stimulus in its RF. Thus, the neuron was neither sensitive to the pre-translation extrafoveal appearance, nor was it sensitive to an intra-translation image change. Error bars denote SEM across trials. The figure's underlying data are included in S5 Data.

signaling of trans-saccadic extrafoveal-versus-foveal image inconsistencies in Figs 2–4 above was contingent upon saccade generation.

Across the population, we reached similar conclusions. Fig 6 shows results of analyses like those in Fig 3, but for the present simulated saccade condition. For each foveated image feature, there were no statistically significant differences between control and image flip conditions ($p = 0.44$ in Fig 6e and $p = 0.055$ in Fig 6f, Wilcoxon signed-rank test in each case; and $p = 0.71$ in Fig 6g and $p = 0.09$ in Fig 6h, Wilcoxon signed-rank test in each case), indicating that the foveal neurons did not "detect" a change from pre-translation extrafoveal image appearance like we saw with real saccades. Consistent with this, there was no significant correlation between the effects of real and simulated saccades in the neurons in which we ran both conditions within the same session (S4 Fig; $r = -0.1034$, $p = 0.4701$ for panel a; $r = -0.1703$, $p = 0.2323$ for panel b). Moreover, the population temporal dynamics (Fig 6a–6d) were consistent with the temporal dynamics shown in the example neuron of Fig 5.

Thus, when foveal SC neurons were visually-stimulated through exogenous means, and not through self-generated saccades, they expectedly only reflected the instantaneous sensory evidence entering into their RF's. Signaling of trans-saccadic inconsistencies between extrafoveal and foveal image appearances by foveal SC neurons was contingent upon saccade generation.

## Extrafoveal stimulus visibility interacts with trans-saccadic error signaling

Because extrafoveal visual acuity is not as good as that in the fovea, including in the SC's retinotopic visual field representation [12,21,40], one should not necessarily expect universally successful signaling of trans-saccadic prediction errors by

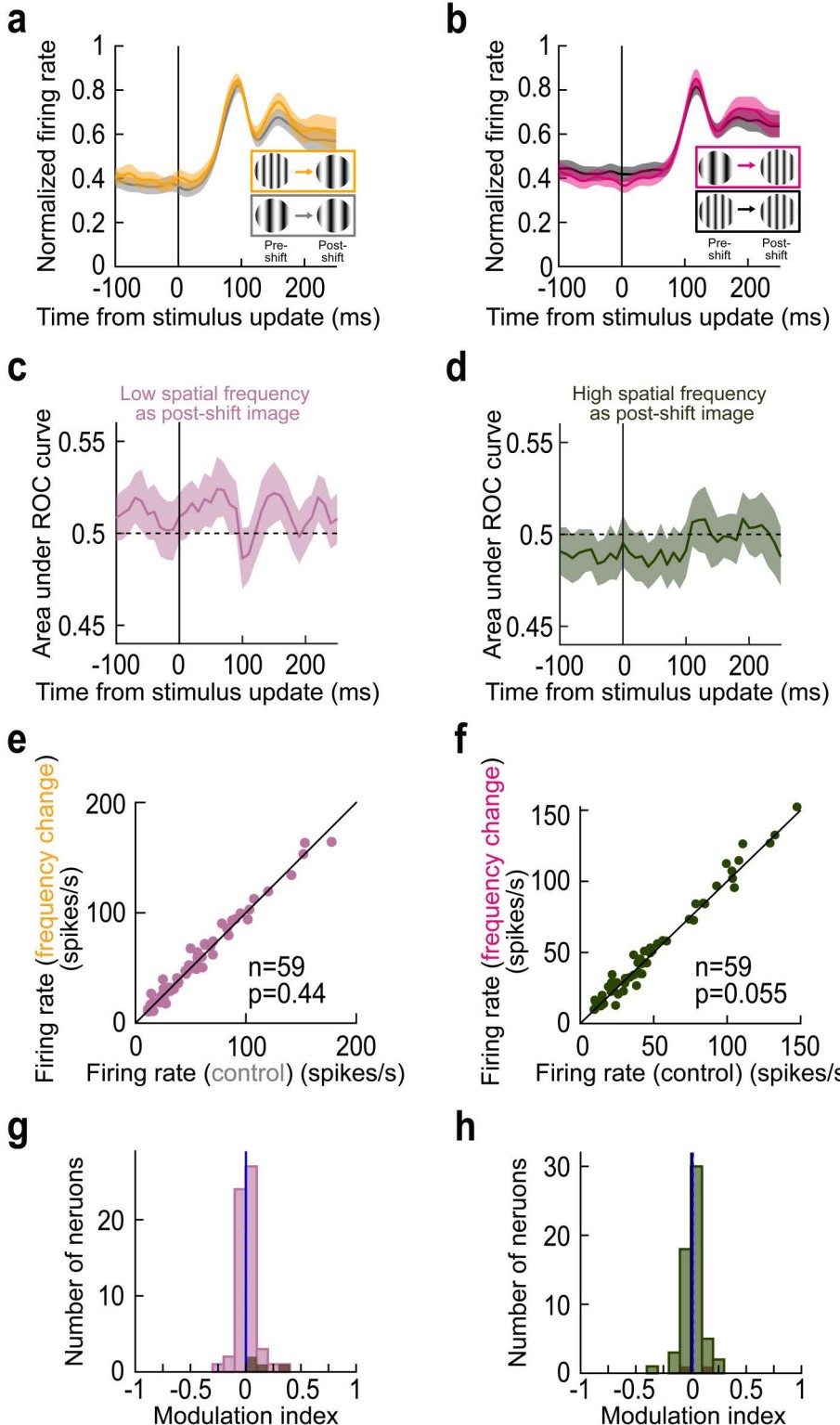

**Fig 6. Lack of foveal neural response elevations in the absence of real saccadic eye movements.** Analyses similar to those in Fig 3, demonstrating a lack of foveal response elevations in the absence of saccades. Error bars denote SEM in a, b, and 95% confidence intervals in c, d. All other conventions are like in Fig 3 above. The figure's underlying data are included in S6 Data.

foveal SC neurons. Specifically, if the relevant extrafoveal visual feature was only subtly changed intra-saccadically, then trans-saccadic elevations of foveal SC reafferent responses might be reduced or eliminated: when extrafoveal visibility is already limited, there is little evidence to allow signaling a trans-saccadic image mismatches.

To test this, we employed object shape changes; rather than changing the spatial frequency of the texture inside the saccade target across saccades, we changed only the outline form encompassing the texture. For either a low or high spatial frequency texture, the pre-saccadic extrafoveal target was a circle, and the post-saccadic foveal target was now a square. These trials were interleaved with all of the other trials of Figs 1–4 ("Methods"). We hypothesized that, with the low spatial frequency texture, extrafoveal target visibility was dominated by the highly salient texture and not by the outline shape (especially for SC neurons preferring low spatial frequencies; ref. [12]). On the other hand, with the less salient fine texture, the form (circle versus square) could be better distinguished extrafoveally, and could, in turn, be sensed by the foveal neurons if it was changed trans-saccadically. Thus, we expected that trans-saccadic signaling of shape changes (between the pre-saccadic extrafoveal and post-saccadic foveal image scenarios) would be weakened with the low spatial frequency texture embedded within the shape.

Fig 7a and 7b shows the results of the same example neuron of Figs 1–2 and 5 under these additional image manipulations. When the shape of the saccade target was changed across saccades, this change was only detected by the foveal SC neuron when the texture inside the shape was a high spatial frequency. We also confirmed this statistically. Across trials, the peak visual reafferent response in the shape-change trials was not different from that in the control trials for the low spatial frequency texture embedded within the saccade targets ($p=0.7587$; Wilcoxon rank sum test; Fig 7a). However, the reafferent response was elevated relative to control in the shape-change trials when the embedded texture was of high spatial frequency ($p=2.67 \times 10^{-4}$; Wilcoxon rank sum test; Fig 7b).

Such an effect was also present across the population (Fig 7c–7f, using similar formatting to earlier figures). When the pre- and post-saccadic targets contained within them a low spatial frequency texture, the median modulation index across the population was 0.0161 (Fig 7e). This was lower than the median modulation index seen in Fig 3g when landing on a low spatial frequency texture (median 0.0541; $p=0.0228$; Wilcoxon signed-rank test comparing the modulation indices in the two conditions). This suggests weakening of trans-saccadic prediction error signaling when the trans-saccadic change was rendered more subtle (also see below for further evidence of such scaling effects). Interestingly, when the embedded texture was of a high spatial frequency, putatively rendering the shape change more visible due to the lower salience of high spatial frequencies, there was no difference in modulation indices between the shape-change trials (Fig 7f) and the spatial-frequency change trials (Fig 3h; $p=0.6530$; Wilcoxon signed-rank test comparing the modulation indices in Figs 3h and 7f). Once again, these results were not an outcome of variability in saccade landing positions (S3b Fig).

Therefore, trans-saccadic signaling of inconsistencies between experienced and expected foveal visual target appearances by SC neurons may fail when the limits on pre-saccadic extrafoveal preview visibility are reached.

## Scaling of error signaling by combined image conflicts

Finally, our shape-change experiments allowed us to next test combined image manipulations. On intra-saccadic image flip trials, we now flipped both the spatial frequency and the encompassing outline form at the same time. This way, we could ask whether signaling of trans-saccadic prediction errors by foveal SC neurons may be viewed as just an "event detector" of any trans-saccadic image inconsistency, or whether the signaling strength might scale with the magnitude of the feature changes generated by the intra-saccadic image flips. From the comparisons made between the neural modulation indices of Figs 3 and 7, we had reason to believe that trans-saccadic prediction error signaling by foveal SC neurons is more than just a simple event detection process.

At the individual neuron level, consistent with the differences between the Figs 3 and 7 population results, we did indeed notice that the modulation of reafferent response strengths on combined feature change trials could be significantly stronger than its modulation on individual feature change trials. For example, Fig 8a and 8b shows the results of the same

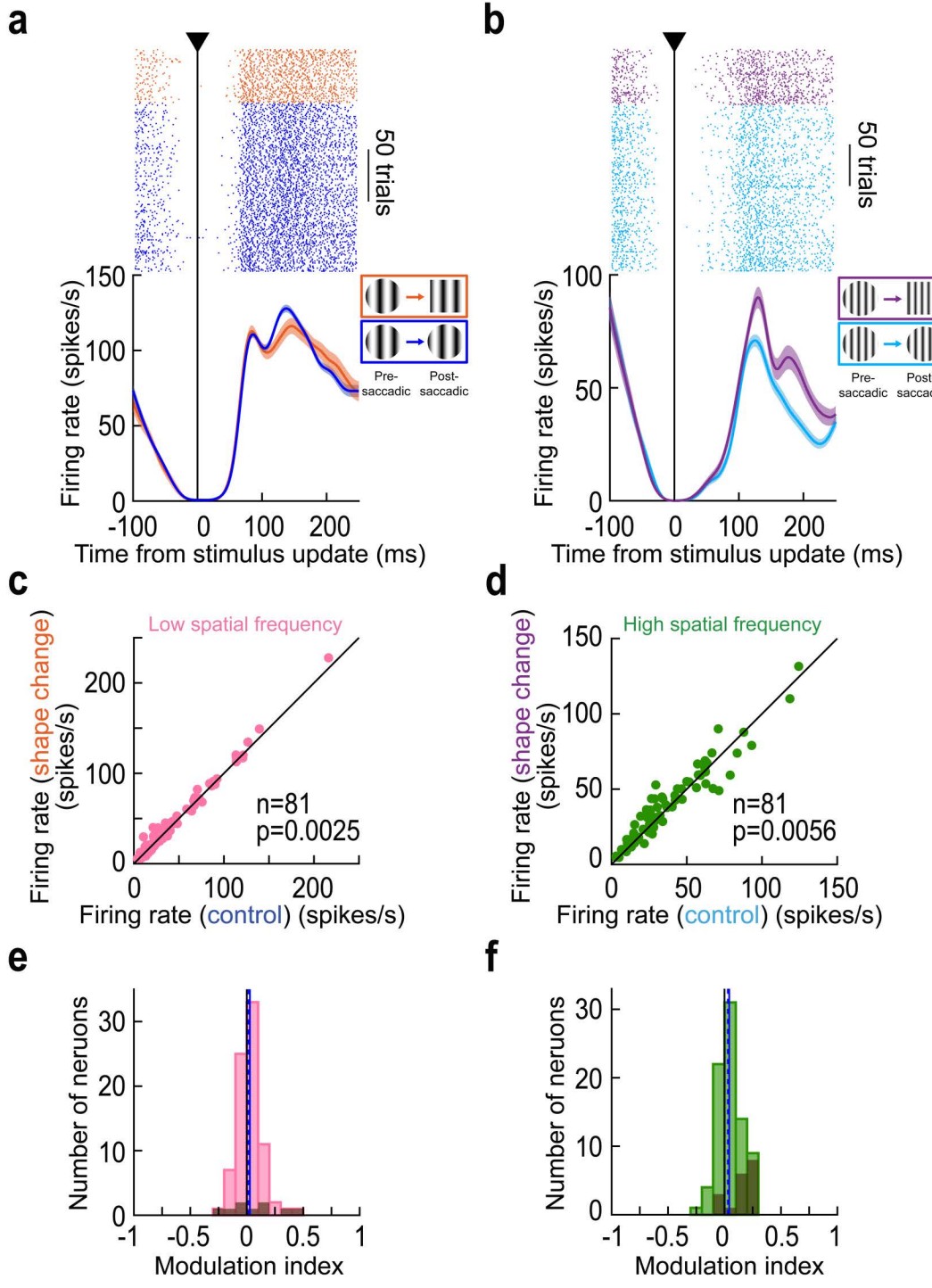

**Fig 7. Generalizability of trans-saccadic error signaling to the case of intra-saccadic outline shape changes.** (a) Visual reafferent responses of the same foveal neuron of Figs 1–2 and 5. Here, we compared the control condition (blue; same as in Figs 1 and 2) of landing on a circular target containing a low spatial frequency texture to the one in which only the outline form was changed (but the foveated texture unaltered). There was no elevation of reafferent response strength by the intra-saccadic image change. We hypothesize that the salient low spatial frequency texture dominated the pre-saccadic visibility of the extrafoveal target, rendering the shape harder to discern pre-saccadically. (b) Consistent with this, when the embedded texture was a high spatial frequency, and thus putatively less salient, a trans-saccadic shape mismatch was successfully detected by the neuron. (c, d) Same as in Fig 3e and 3f, but for the shape-change manipulation. Across the population, there was a significant elevation in both cases ($p = 0.0025$ in c and $p = 0.0056$ in d; Wilcoxon signed-rank test). (e, f) For the neural modulation indices, there was also a positive population index in both cases ($p = 0.03$

in e and $p=0.0016$ in f; Wilcoxon signed-rank test). However, as mentioned in the text, the modulation indices in e were significantly weaker than those in Fig 3g, suggesting a weaker effect of the shape changes when the embedded textures were of low spatial frequency. All conventions are similar to those in Figs 2 and 3. Also see S5a, S5b, S5e and S5f Fig for population ROC and neural firing rate dynamic analyses, supporting the results in this figure. The figure's underlying data are included in S7 Data.

neuron as in Figs 1, 2, 5 and 7 above, but now with both a shape and spatial frequency change across saccades. When saccades landed on a low spatial frequency texture (plus a changed shape; Fig 8a), there was an elevated reafferent response relative to control, and this elevation was larger in magnitude than that observed on either the spatial-frequency (Fig 2b) or shape (Fig 7a) change trials alone: the difference in peak reafferent response from control was 26.82 spikes/s in the combined change trials, whereas it was 16.11 and 1.02 spikes/s in Figs 2b and 7a, respectively. When the saccades landed on a high spatial frequency target instead, and again in the combined change trials (Fig 8b), the reafferent response elevation was larger than that observed on the shape-only trials (Fig 7b; 38.74 versus 19.01 spikes/s), and also more robust than that observed for the spatial frequency trials alone (Fig 2c; 38.74 versus 27.06 spikes/s). Thus, combining multiple intra-saccadically altered image features strengthened the elevations of reafferent response magnitudes caused by inconsistencies between the pre- and post-saccadic visual image appearances.

Next, after confirming that the combined image feature inconsistencies still caused trans-saccadic error signaling in our population (Figs 8c–8f, S5c, S5d, S5g and S5h), we tested to what extent the neural modulation indices on combined (shape plus spatial frequency) image trials approached the sum of the neural modulation indices obtained in the individual image manipulations separately. Specifically, our control trials were the same for both types of experiments, since these control conditions constituted only circular saccade targets and a single spatial frequency ("Methods"). Thus, on trials with only spatial frequency flips or only shape flips, the control condition that we were comparing to was the same. This allowed us to compute neural modulation indices on the trials with combined image changes and see whether they approached the sum of the individual modulation indices.

For each neuron, we plotted, on the x-axis, the sum of neural modulation indices obtained from the modulation indices of the individual image manipulations (shown in Figs 3 and 7). Then, on the y-axis, we plotted the actual (measured) neural modulation indices obtained from the combined image manipulations (Fig 8e and 8f). There was indeed a clear positive correlation, both when landing on a low (Fig 9a; $r=0.6632$; $p<0.000001$) and high (Fig 9b; $r=0.8521$; $p<0.000001$) spatial frequency texture. Therefore, signaling of trans-saccadic prediction error by foveal SC neurons does scale (albeit sub-additively when landing on a low spatial frequency texture) with the amount of feature conflicts experienced by the neurons.

We should also emphasize here that we additionally ran the simulated saccade experiments, on the same neurons as in Fig 6, with the shape-only and combined (shape plus spatial frequency) change experiments. In both cases, we did not observe systematic elevations in sensory responses to the afferent signals received by the translating images into the neurons' RF's. This can be seen in S6 Fig.

## Discussion

In this study, we investigated the activity of foveal SC neurons [21] as these neurons were visually stimulated by saccade-induced translations of extrafoveal visual stimuli into their RF's. We tested the hypothesis that foveal SC neurons can receive, trans-saccadically, an expected appearance of the foveated saccade target (based on this target's pre-saccadic visual features). We picked only neurons that were not visually stimulated by the extrafoveal saccade target. Then, during the saccade, we altered the target's appearance, such that the post-saccadic foveated image feature (now visually driving the neurons) was different from that expected from the pre-saccadic visual property. We found that visual reafferent responses were systematically elevated if there was a trans-saccadic mismatch in saccade target appearance. This elevation did not happen with simulated saccades, and it also scaled with the extent of feature mismatches that we introduced across saccades.

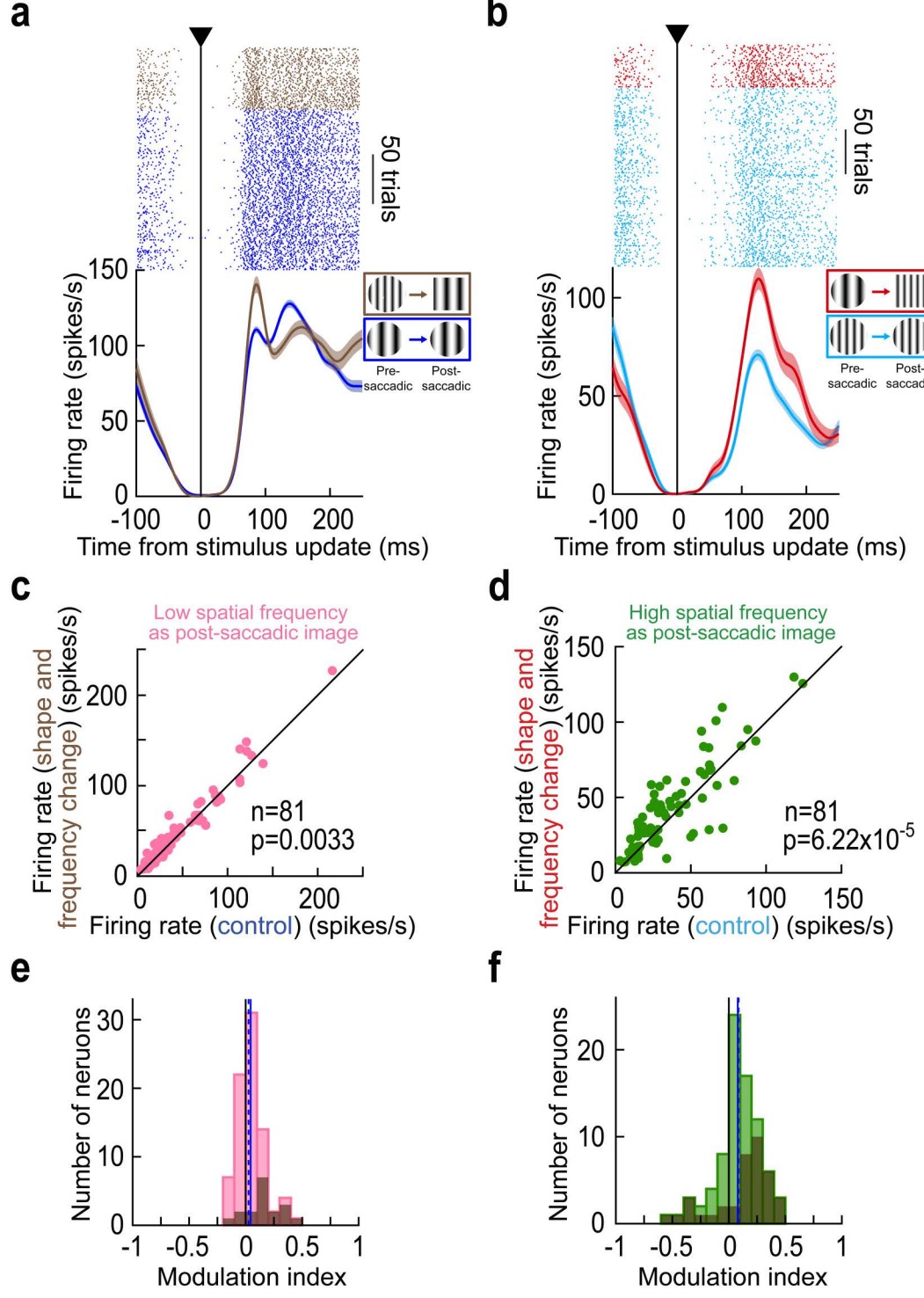

**Fig 8. Scaling of trans-saccadic prediction error signaling by foveal SC neurons with combined intra-saccadic feature changes.** (a, b) Results from the same example neuron of Figs 1, 2, 5 and 7. Here, relative to the same control trials of Figs 1 and 2, we checked reafferent responses when both the outline shape and the embedded spatial frequency texture within it were altered intra-saccadically. Both when landing on a low (a) or high (b) spatial frequency texture, the reafferent responses of the neuron were elevated more strongly than either for only the spatial frequency (Fig 2) or for only

the shape (Fig 7) change trials. (c, d) Consistent trans-saccadic error elevations in foveal neuron reafferent responses were observed across the population ($p = 0.0033$ for c and $p = 6.22 \times 10^{-5}$ for d; Wilcoxon signed-rank test). (e, f) Similarly, population modulation indices were also positive ($p = 0.0023$ for e and $p = 6.6 \times 10^{-5}$ for f; Wilcoxon signed-rank test). Error bars denote SEM, and all conventions are the same as in earlier figures. Also see S5c, S5d, S5g, and S5h Fig for population ROC and neural firing rate dynamic analyses, supporting the results in this figure. The figure's underlying data are included in S8 Data.

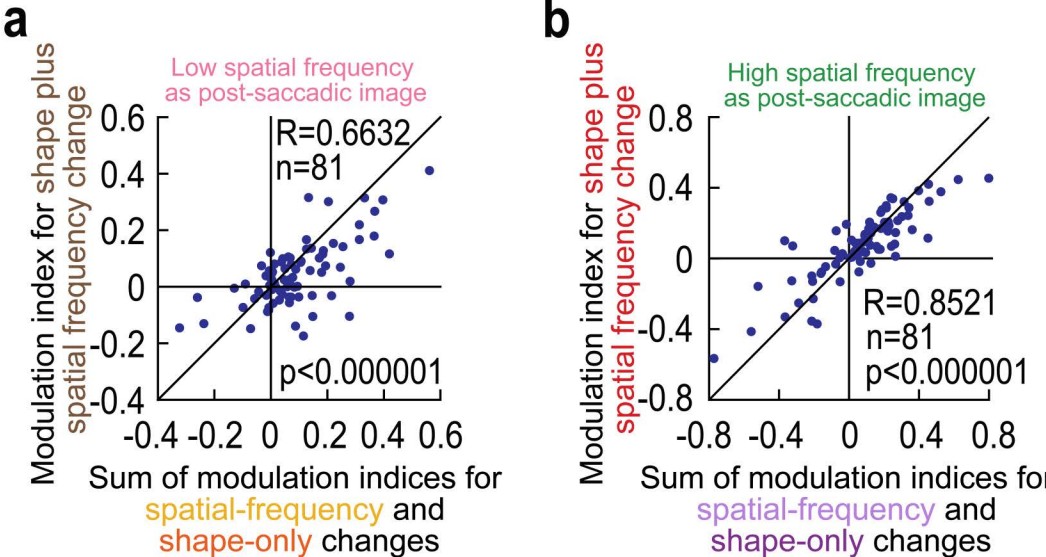

**Fig 9. Scaling of trans-saccadic error signaling in the foveal SC by combined intra-saccadic image changes.** (a, b) When landing on either a low (a) or high (b) spatial frequency texture, we computed on the *x*-axis the sum of modulation indices obtained from the experimental trials with only spatial frequency intra-saccadic changes (Fig 3) and only shape changes (Fig 7). Then, on the *y*-axis, we calculated the actual measured modulation indices from the trials containing both spatial frequency and shape intra-saccadic image updates. In both cases, there was a clear positive correlation, suggesting a scaling of prediction error signaling effects in the foveal SC with the magnitude of feature inconsistencies across saccades (correlation coefficients are indicated in each panel; $p < 0.000001$ in each case). The figure's underlying data are included in S9 Data.

We interpret our results within the framework of predictive coding [7,8,33,34]. In this framework, neural responses would be elevated under so-called prediction error situations. In our case, a stable saccade target across saccades is the most expected sensory consequence of eye movements (i.e., having no prediction error). This is because the real world is generally stable across rapid eye movements [41,42], despite translations of its image on the retina. Thus, experiencing a different foveal image from that expected based on the pre-saccadic extrafoveal saccade-target appearance constitutes a trans-saccadic prediction error; accordingly, the visual environment can be interpreted as having been unstable across the saccade.

Consistent with our interpretation, some of the oldest ideas about perceptual stability in the face of eye movements have revolved around concepts of "template matching": a pre-movement template (or prediction) is formed, and post-saccadic perceptual stability is established when a post-movement image matches the pre-emptively formed template [3,9]. Neurobiologically, changes in visual RF shapes and locations do indeed occur peri-saccadically [43–50]. In sensory-motor areas, such as frontal eye field (FEF), lateral intraparietal area (LIP), and SC, these RF changes have long been termed "predictive remapping" [43,44,50]. However, while the exact details of spatial updating continue to be under scrutiny [46,48,49,51,52], such spatial updating remains significantly and conceptually different from our focus here: how is visual sensation of a stably present visual object in the environment handled when an eye movement brings this object

inside a given neuron's RF? In particular, how are visual feature representations, not just retinotopic spatial representations, handled across saccades?

One aspect of this question was already addressed previously by Crapse and Sommer [34], albeit again with a focus on spatial location. These authors recorded from extrafoveal FEF neurons. They presented a visual stimulus that was stable and brought into the visual RF of an FEF neuron by a saccade. When the stimulus remained stable across the saccade, the visual reafferent response to the stimulus (as it entered into the RF of the recorded neuron) was not as strong as when the same stimulus was experimentally translated intra-saccadically (such that it still occupied the same post-saccadic RF location). Thus, the recorded FEF neuron "signaled" (by elevating its firing rate) a deviation from the "predicted" location of a stable stimulus, even though the stimulus was outside its RF pre-saccadically; the neuron, therefore, detected the unexpected intra-saccadic change in stimulus position. We do not know of such effects in the SC, and certainly not with visual features other than spatial location. Thus, we attempted to fill this gap here, and we also used target features, rather than location, as the manipulation of interest for us. Importantly, we additionally did this in the foveal visual processing domain, which is, in any case, one of the most important aspects of active visual scanning by saccadic eye movements. Our results, in this sense, support the recent experiments demonstrating that foveal perception is influenced, right before saccade onset, by extrafoveal saccade target appearance [53,54]. These results are also in line with previously described peripheral preview effects; in such effects, changes in both behavioral performance (such as reaction times) and cortical brain activity have been observed if the post-saccadic foveated image was rendered different from the pre-saccadic preview [4,55–57].

Having said that, we do not wish to imply a complete dissociation of our effects from predictive spatial remapping models. We believe that accounts of predictive remapping of spatial representations that include remapping of visual feature information to the fovea likely contribute to our observations [53,57–61]. In that regard, we find it intriguing that we observed evidence of a kind of "capacity limit" to the utility of pre-saccadic previews for our foveal SC neurons to signal trans-saccadic prediction errors. For example, our neural modulation effects scaled with the amount of intra-saccadic feature changes that we introduced (e.g., Figs 8 and 9) and also with the putative pre-saccadic extrafoveal visibility of the saccade target (e.g., Fig 7). This makes sense. In fact, spatial resolution changes strongly with retinotopic eccentricity, and SC spatial frequency tuning curves also exhibit a gradient of decreasing preferred spatial frequency with increasing eccentricity [12]. Thus, if the extrafoveal preview is limited in its information, then so is the ability to detect intra-saccadic feature changes from this preview. This is not necessarily a disadvantage for trans-saccadic integration; rather, it confers the visual system with some tolerance to incidental retinotopic image changes. Otherwise, prediction errors might be signaled by the brain all too often and thus disrupt visual processing as opposed to supporting perceptual stability. Interestingly, the limits of such a tolerance window of the visual system might have been successfully exploited by the famous Mona Lisa painting [62]. Specifically, when viewed peripherally, the Mona Lisa's mouth is perceived as smiling more than when viewed foveally; low-pass filtering of the painting also revealed a smiling mouth, suggesting that the peripheral percept is mediated by spatial frequency [62].

The above example suggests that it would be interesting in the future to test foveal SC neurons with faces and other object stimuli as the saccade targets. Indeed, the (extrafoveal) SC detects faces [13,15,63–66], and real-life objects in general [13], in its very first visual responses. Moreover, the (extrafoveal) SC senses object stimuli already in its saccade-related motor bursts [16]. Thus, SC neurons may already have a peripheral "object" preview both at stimulus onset and at saccade generation. Testing foveal SC neurons with these ecologically-relevant stimuli across saccades would then allow direct translations of cortical peripheral preview effects [55,56] to relevant subcortical brain areas, and to the fovea.

The concept of remapping also invites mechanistic studies of how rerouting of information happens across saccades. It has been suggested that such rerouting might be facilitated by oscillatory coherence between retinotopic cortical sites that would experience the pre- and post-saccadic image [67], and it was also found that saccades might trigger cortical

traveling waves [68]. These ideas are interesting to revisit from an SC perspective especially because classic SC studies have indeed revealed wave-like spreading of SC activity across saccades [69,70]. In fact, such SC spreading might lead cortical waves in time, and therefore be more relevant for the predictive mechanisms suggested by our work.

Our choice to study the SC was also motivated by additional factors. For example, besides integrating visual evidence from the entire visual system [11], the SC issues saccade motor commands [20]. Importantly, and as just mentioned, at the time of saccade generation, such integrated visual evidence gets reactivated in the SC, exactly when predictions might be relayed to foveal visual representations according to our framework: SC saccade-related motor bursts embed within them a sensory representation of the saccade target's visual properties [16,19]. Thus, our foveal neurons might receive such evidence from the extrafoveal motor bursts of the SC. It would be interesting in the future to explore whether such information transfer happens intra-collicularly or through loops outside of the SC, especially because it is also known that an influence in the reverse direction (from the fovea to the periphery) additionally occurs [71]. This could be done, for example, by reversibly inactivating the extrafoveal SC representation and checking whether trans-saccadic error signal by foveal neurons would survive.

We also think that the SC is interesting to study because of its importance for signaling salience. Classically, such salience has been described in the image domain. That is, when spatiotemporal movies are played to the retina, then the SC can signal salient "locations" in retinotopic images at specific time snapshots [22–25]. However, salience can also be represented in the temporal domain. For example, our results indicate that foveal SC neurons can signal salient post-saccadic events highlighting a "failure" of expected visual scene stability as seen by the very same neurons. Thus, the same retinotopic location responds differentially at different times. We believe that this idea can also extend to non-saccadic situations. For example, if there are known temporal contingencies in the environment with certain types of stimuli, then we can expect that SC neurons can still be sensitive to violations in these contingencies even in the absence of saccades. This can happen, say, if certain object motions follow strict or systematic physical rules to their sequencing, such as in the case of biological motion. If the SC does indeed behave as a temporal salience detector for detecting violations in temporal contiguity across saccades, then this can be part of a larger framework for building invariant object detection in the cortex [72,73].

We now turn to potential limitations of our study and its interpretations. For one, it could be argued that our elevated reafferent responses reflect a relief from adaptation. That is, in the control trials, the pre-saccadic image was the same as the post-saccadic one, whereas it was different on the image-flip trials. However, because the SC is retinotopic, adaptation in our experiments would have been specific to the extrafoveal neurons that saw the pre-saccadic stimulus and not to our recorded foveal neurons. It is not clear why foveal SC neurons should reflect the adaptation state of extrafoveal ones. This possibility could potentially be tested in the future by exploring neural trajectories in the population manifold represented by foveal SC neurons, both at the time of extrafoveal stimulus presentation and also around saccade onset. For example, population manifolds of extrafoveal SC neurons are orthogonal to each other between the stimulus and saccade generation epochs [16], suggesting that such an approach may reveal more than just potential across-SC adaptation effects. More importantly, if adaptation was indeed present, then this in itself would constitute a trans-saccadic transfer of information from the extrafoveal to the foveal SC. This would be directly consistent with our interpretation that foveal SC may receive information about the state of extrafoveal scene appearances before saccades.

Our simulated saccades were also not direct replicas of real saccadic trajectories. Thus, it might be argued that our simulated saccades were not "sufficient" to cause elevated responses in image-flip trials. However, our experience with these types of simulated saccades from earlier experiments is that they often lead to stronger, rather than weaker, modulations [74]. Also, the fact that we did not see pauses in activity before the visual responses (Fig 5) supports our view that the neurons were not stimulated before the image translations, and that they simply passively received new stimuli entering their RF's. Thus, it is not too surprising that they might not detect a pre-translation image difference.

It would also be interesting in the future to have a perceptual readout from our paradigm. That is, here, we did not query whether the monkeys experienced perceptual instability or not. This would be important to establish, and we already have interesting hints from our current data. Specifically, when we analyzed the timing of the first catch-up saccade after the primary saccades in our data, we found that it was sensitive to intra-saccadic stimulus changes, especially when landing on a high spatial frequency texture (S7 Fig); the effect was consistent with a slight delaying of catch-up saccades on prediction error trials, which is the expected consequence of elevated foveal SC activity. This motivates further searches of perceptual readouts of trans-saccadic integration in our paradigm, along with causal manipulations of SC activity, in order to establish whether prediction error signaling has direct influences on perceptual experience. Given the role of the SC in attention [75], we suspect that our neural modulations would be relevant for altering behavior. For example, we could inactivate the extrafoveal SC and ask whether trans-saccadic error signaling by foveal SC neurons would survive, and whether perceptual readouts can also be affected.

Finally, we think that SC neurons with large RF's can potentially contribute to the transfer of information from the extrafoveal neurons at the time of saccade onset [16] to the foveal neurons post-saccadically, exactly for realizing prediction error signaling. In our analyses, we strictly excluded such neurons ("Methods"). However, we had plenty of such excluded neurons with subtle elevations in their activity at extrafoveal stimulus onset. That is, their peri-foveal RF's extended out to be marginally driven by extrafoveal stimulus onset. When we analyzed those excluded neurons like we did in this study, we found that they had trans-saccadic modulations that were not too different from those reported here (for example, like the reafferent responses of Figs 1 and 2). Thus, intermediate RF's spanning both pre- and post-saccadic visual representations could be critical for the mechanisms explaining our results, and they could exist through wide-field synaptic connections in the SC [76].

## Methods

### Experimental animals and ethical approvals

We collected data from two adult, male rhesus macaque (*Macaca mulatta*) monkeys (A and M), aged 8–9 years.

The animals were prepared previously for behavioral and neurophysiological procedures. They each had a head holder for stabilization of head position during the experiments, scleral search coils for eye tracking, and a recording chamber with craniotomy for access to the superior colliculus (SC).

All experiments were approved by ethical committees at the regional governmental offices of Tübingen (Regierungspräsidium Tübingen), under licenses CIN 3/13 and CIN 4/19G, and the experiments complied with all guidelines and regulations.

### Laboratory setup

The experiments were conducted using the same setup as in our earlier studies [13]. Specifically, we used a custom-built modification to PLDAPS [77], which employed the PsychToolbox [78–80] for display control. The system synchronized with the OmniPlex system from Plexon for data acquisition and storage.

The use of the PsychToolbox allowed us to synchronize our experimental control system with the graphics card controlling our display. This enabled direct access to the vertical blanking signal of each frame update, which was important for achieving gaze-contingent intra-saccadic display changes in some trials (see details below). In previous studies performed during the earlier years of the laboratory's existence [32,81–83], we performed extensive tests of this system with external photodiodes detecting every single frame update. This confirmed that we successfully synchronized our experimental control system with each frame command. Consistent with this, the reafferent responses after saccades (e.g., Figs 1, 2, 3a and 3b) had very similar timing across trials, regardless of whether there was an intra-saccadic display change or not (also see Fig 2a for display timing relative to saccades, and S2 Fig for other eye movement parameters with and without intra-saccadic display changes).

We used a linearized and calibrated CRT display monitor with a refresh rate of 85 Hz and approximately 34 pixels/deg resolution. The monkeys sat approximately 72 cm from the display, and experiments were conducted with a gray background on the display (having 26.11 cd/m$^2$ luminance). White stimuli (for example, for a fixation spot) had a luminance of 79.9 cd/m$^2$. We tracked eye movements using the magnetic induction technique [84,85], and we recorded neural activity using 16- or 24-channel V-Probes from Plexon.

## Experimental procedures

For response field (RF) mapping, we employed a fixation task. The monkeys maintained fixation on a central fixation spot (black), and we then presented a small white spot at pseudorandom locations across trials [86]. The fixation and target spots were each approximately 10.8 × 10.8 min arc in size. We mapped RF's to ensure that there were no visual responses to the extrafoveal visual stimulus onset in the main tasks (e.g., Fig 1b).

For the main saccade task, the monkeys fixated a central (white) fixation spot of the same dimensions as above. After 300–700 ms from foveating the fixation spot, we presented the extrafoveal stimulus. This stimulus appeared at 8 or 10 deg to the right or left of the fixation spot in all sessions except two; for these two sessions, we placed the stimulus at ±5.5 deg horizontally and −4.7 deg vertically, or ±4.6 deg horizontally and −4.4 deg vertically. This placement was dictated by other unrelated experiments that we were simultaneously performing during these two sessions in the primary visual cortex. The extrafoveal stimulus was a circle of 3 deg radius. Inside, there could be a vertical sine wave texture of high contrast (100%), having either 1 cycle/deg (cpd) or 4 cycles/deg (cpd) spatial frequency. We called the former the low spatial frequency texture and the latter the high spatial frequency texture. After a delay period of 500–1,000 ms, we removed the fixation spot, instructing the monkeys to generate a saccade towards the extrafoveal stimulus.

When eye position exited a virtual window surrounding the fixation spot, the computer flagged a saccade detection event. This triggered a stimulus update on the very next graphics display frame update. In control trials, the initial shape was always a circle (with either low or high spatial frequency embedded within it), and the display update event was fictive. That is, the stimulus was unchanged in this case (of course, the display itself was always being refreshed from frame to frame, as expected from such display technology). Such control trials constituted 50% of all trials within a session. The remaining trials were split among the different intra-saccadic change conditions. In one case, the initial stimulus was a circle and the final stimulus was still a circle, but containing the other spatial frequency. That is, if the initial circle encompassed 1 cpd, then the changed stimulus had 4 cpd. These spatial-frequency change trials constituted 16.67% of all trials. In another case, the initial shape was a circle containing either the low or spatial frequency texture, and the final shape was a square (6 × 6 deg in size) with the same texture. These shape-change trials constituted 16.67% of all trials in a session. Finally, there were trials with a circle initial shape; the final shape was a square (6 × 6 deg) containing a different spatial frequency (shape plus spatial frequency trials). These constituted 16.67% of all trials in the session. Note that the stimulus never moved in any of the trials. It was the eye that was moving and causing the stimulus to move towards the fovea on the retina. After the foveating saccade to the extrafoveal stimulus, the monkeys maintained fixation for another 500 ms before being rewarded with juice droplets. We collected 322–1,440 trials of this task per session.

In some sessions, we also ran a simulated saccade version of the above paradigm. Here, the monkey always maintained fixation. Instead of a go signal to generate a saccade (removal of the fixation spot), we simply translated the extrafoveal stimulus towards the center of the display. The translation proceeded as follows: the target at the initial 8 deg eccentricity was translated to 5.5 deg in the first frame (i.e., 2.5 deg in 11.76 ms), and then to 1.5 deg in the second frame (i.e., a shift of 4 deg in 11.76 ms), and finally to the final central position in the next frame (i.e., a shift of 1.5 deg in 11.76 ms). Thus, the simulated saccades had an accelerating and decelerating profile with a peak speed of 340 deg/s, which is similar to our real saccades' peak speeds (e.g., Fig 2a). Image-change trials involved changing the relevant image feature in the first frame of the translation. The experiment was otherwise the same as the real saccade version.

We did not run the simulated saccade experiment during the two sessions in which the saccade target was at an oblique position relative to the fixation spot. We collected 647–841 trials of this task per session.

## Data analysis

We detected all microsaccades and saccades using our established methods [32,87]. In the saccade task, we only accepted trials in which saccades landed within a radius of <2 deg from the center of the saccade target. This ensured that the saccade target activated our foveal neurons post-saccadically (S3 Fig).

We sorted neurons offline using Kilosort [88]. In total, we had 249 neurons in our database. However, we excluded neurons in which there was an elevation (however small) in firing rates after extrafoveal stimulus onset in our main task (relative to pre-stimulus firing rates). This is because we wanted to exclude any remote possibility that our foveal neurons were visually-stimulated by the pre-saccadic extrafoveal stimulus. We also confirmed this with our RF mapping task, as well as by measuring post-stimulus activity in our accepted neurons (S1 Fig). We also excluded neurons in which the average firing rate in the interval 0–100 ms from fictive stimulus update time (i.e., in control trials) was less than 5 spikes/s, to ensure that we had a proper post-saccadic visual response to analyze. This left us with 81 neurons, from 20 sessions (out of 23 total), that we included in the final database.

To check for proper intra-saccadic stimulus updates, we collected all saccades and related their parameters (onset, peak velocity, and end times) to the time after stimulus update completion (S2a Fig). We also measured eye speed at the time of stimulus update completion (Figs 2a and S2b), in order to confirm that we had as much stimulus motion blur on the retina as possible at the time of intra-saccadic target changes.

We checked other behavioral metrics of the saccades as well. For example, we plotted radial eye speeds with and without intra-saccadic stimulus changes (S2c and S2d Fig), as well as radial eye displacements (S2e and S2f Fig). The former plotting allowed us to check whether post-saccadic drifts [32] were altered by intra-saccadic stimulus changes, and the former and latter plotting allowed us to investigate whether saccades were truncated [89] or otherwise altered by the intra-saccadic changes. Finally, we measured the latency of the first catch-up saccade after the primary saccade (S7 Fig). This allowed us to check whether elevations in reafferent responses in foveal SC neurons (on intra-saccadic change trials) altered subsequent saccade timing.

To measure reafferent response strength, we calculated the peak firing rate in the interval after the end of stimulus updates. For low spatial frequency foveated targets, the interval was 0–120 ms from real or fictive stimulus update; for high spatial frequency foveated targets, it was 0–150 ms. The two different intervals reflected the fact that responses were delayed for high spatial frequency foveated stimuli [12,29]. For the simulated saccade version of the task, we used the same measurement intervals relative to the stimulus update times. Note that in all cases, we focused only on the initial reafferent response and not also later modulations in firing rate that could happen. This is because such later modulations coincided with potential catch-up saccades, which (due to their small size) were expected to significantly alter foveal SC activity. Also note that we used stimulus update time as the alignment point for measuring our reafferent responses. Because we had a fictive update time stamp also for the control trials, and because online saccade detection was quite repeatable across trials (e.g., Fig 2a), this meant that we had similar alignment of firing rates for both the control and intra-saccadic stimulus change trials. This, in turn, allowed for a fair comparison between the conditions. In addition, the stimulus update time was always at a time of maximal suppression of neural activity (e.g., Figs 2b, 2c, 3a, 3b, 7a, 7b, 8a, and 8b), which was a suitable point of temporal reference. Having said that, we also tried all analyses after aligning to saccade end instead, and we reached the very same conclusions. Firing rate curves looked shifted in time, confirming the repeatable nature of our saccade-contingent stimulus updates. Our choice to use stimulus update times made it easier for us to also use this time in the cases in which no real saccades were generated.

Our modulation indices were calculated for each neuron as follows: firing rate on the intra-saccadic change condition of interest minus firing rate on the corresponding control condition, divided by the sum of the firing rates.

We performed statistical analyses in two ways. First, across neurons, we measured the firing rate on the intra-saccadic change of interest and the firing rate on the corresponding control condition. We then performed a Wilcoxon signed-rank test across neurons. Second, we checked whether the population modulation indices were different from zero using a Wilcoxon signed-rank test. For all depictions of population modulation indices, we also highlighted the neurons that were individually significant in a given analysis. Individual significance was assessed by comparing the firing rate measures of the population of control trials to the firing rate measures of the population of trials in a given intra-saccadic stimulus change condition, using a permutation test across trials (with 10,000 repetitions).

Finally, we also depicted the neural population dynamics in two additional ways. In one, we plotted the average normalized population firing rate curve in a given condition. To do so, we obtained the average firing rate curve of each neuron and each condition (across trials of the same condition). Then, we normalized each average firing rate curve of the neuron by the peak of the average firing rate curve of the corresponding control condition (low-to-low spatial frequency when landing on a low spatial frequency texture, or high-to-high spatial frequency when landing on a high spatial frequency texture). Then, we averaged across neurons and depicted mean and SEM curves. In the second approach, we performed receiver-operating-characteristic (ROC) analyses, like we recently did [16]. Here, we were testing the discrimination performance between a stimulus change condition and a control condition as a function of time relative to the stimulus change. If reafferent responses were elevated on intra-saccadic change trials, then the area under the ROC curve was expected to elevate relative to pre-saccadic periods. All of our area under the ROC curve plots included 95% confidence intervals around them.

## Supporting information

**S1 Fig. Lack of visually-evoked bursts in the foveal neurons at the time of extrafoveal stimulus onsets. (a)** For all neurons included in our main analyses of Figs 1–4 and 7–9, we measured neural activity 50–150 ms after extrafoveal stimulus onset (during maintained gaze fixation), and we compared it to pre-stimulus activity. Here, the appearing stimulus was a circular patch containing a low spatial frequency texture. There was a reduction in activity, rather than visual bursts. **(b)** Similar observations for the case in which the extrafoveal stimulus had a high spatial frequency texture embedded within it. The figure's underlying data are included in S10 Data.
(PDF)

**S2 Fig. Confirmation of intra-saccadic stimulus updates, as well as confirmation of a lack of truncation of primary saccades. (a)** For all trials containing a real or fictive image update (from the basic paradigm of spatial frequency changes and circular outline shapes of the saccade targets), we measured (also across all sessions) the times of saccade onset, saccade peak velocity, and saccade end relative to when the image flip event was completed. We always had intra-saccadic image updates in our experiments. **(b)** Importantly, the eye speed was always high by the end of the image update. Here, for each session, we calculated the average eye speed by the end of the image update event, and we plotted a histogram across sessions. For the great majority of sessions, the image speeds on the retina by the end of intra-saccadic image updates were >150 deg/s. For the two sessions with speeds of approximately 50 deg/s, the saccade targets were placed a bit closer to the fovea (thus having smaller saccades) because the sessions involved other experiments involving the primary visual cortex (we thus tailored the target locations to the cortical neurons explored during the same sessions for other purposes). However, eye speed was still relatively large during these outlier sessions. **(c, d)** For the main conditions of Figs 2 and 3 we plotted radial eye velocity on control and intra-saccadic stimulus change trials. Each plot shows the average velocity curve across sessions (with horizontal saccade target locations, which were the majority), and the error bars show SEM across sessions. The saccadic profiles were not affected by intra-saccadic display updates, including post-saccadic drift speeds. This means that the saccades were not truncated by the stimulus changes. Similar observations were made for the few oblique sessions. **(e, f)** Similar to **c**, **d** but now plotting radial saccade

amplitude across sessions. There was again no evidence of saccade truncation by the intra-saccadic stimulus changes. Similar conclusions were reached for all other experiments (e.g., with shape changes). The figure's underlying data are included in S11 Data.
(PDF)

**S3 Fig. Confirmation of post-saccadic visual stimulation of our foveal neurons by the saccade targets. (a)** Across trials and sessions from our basic paradigm, we plotted the retinotopic position of the saccade-target patch (circles) upon saccade end. There was variability due to variability in saccade metrics. However, the foveated image patches were always encompassing our recorded foveal neurons (dark gray spots indicating the foveal RF hotspot locations). Thus, we always had robust visual reafferent responses. Note also that such jitter in saccade landing should have weakened reaf-ferent responses in some cases, rather than strengthened them, because it could cause visual stimulation at sub-optimal RF positions. This suggests that our main results (dominated by elevated reafferent responses) are not explained by jitter in saccade landing positions. **(b)** Similar observations from the shape-change trials (Methods) of our paradigm. Here, the foveated stimulus had a square outline instead of a circular one. The figure's underlying data are included in S12 Data.
(PDF)

**S4 Fig. Lack of relationship between the modulation indices associated with real versus simulated saccades. (a)** For a low spatial frequency foveated patch after either real or simulated saccades, we plotted the modulation indices from Fig 6g on the *x*-axis (for simulated saccades) and those from Fig 3g on the *y*-axis (for real saccades). The shown neurons are those for which we ran both conditions together in the same session. In the simulated saccade case (*x*-axis), the modulation indices straddled zero; however, in the real saccade case (*y*-axis), the modulation indices were largely positive. Thus, there was no correlation between the two situations ($p = 0.4701$). **(b)** Similar observations for a high spatial frequency foveated image ($p = 0.2323$). The figure's underlying data are included in S13 Data.
(PDF)

**S5 Fig. Population firing rate dynamics in the shape change and shape and frequency change conditions. (a, b)** ROC analyses for the experiments of Fig 7. Whether landing on a low or high spatial frequency texture, there was an elevation in area under the ROC curve in the reafferent response epoch. **(c, d)** The effects were larger for the combined shape and spatial frequency change trials (Fig 8). **(e, f)** Population firing rate dynamics for the experiments of Fig 7. **(g, h)** Population firing rate dynamics for the experiments of Fig 8. Error bars denote 95% confidence intervals for **a**–**d** and SEM for **e**–**h**. The figure's underlying data are included in S14 Data.
(PDF)

**S6 Fig. Confirmation that shape-change and shape plus spatial frequency changes had much weaker effects on foveal SC neurons during simulated saccades than during our main experimental manipulations with real sac-cades. (a, b)** Population results like in Fig 6e and 6f, but now for the shape change trials and a low (**a**) or high (**b**) spatial frequency post-translation texture. There were weaker differences in visual response strength relative to control than with real saccades. **(c, d)** Similar observations for the shape plus spatial frequency change trials. **(e, f)** Neural modulation indi-ces for **a**, **b** ($p = 0.8$ and 0.049 for **e** and **f**, respectively; Wilcoxon signed-rank test). **(g, h)** Neural modulation indices for **c**, **d** ($p = 0.22$ and 0.40 for **g** and **h**, respectively; Wilcoxon signed-rank test). The figure's underlying data are included in S15 Data.
(PDF)

**S7 Fig. Influence of the neural modulations in this study on the timing of catch-up saccades after the primary eye movements. (a)** For the basic experiments of Figs 2 and 3, and when landing on a low spatial frequency texture, the distribution of catch-up saccade times (for the first saccade after foveating the target) was very slightly skewed towards longer latencies on intra-saccadic change trials, but there were also saccades with very short latency. **(b, c)** Similar

observations for the shape (**b**) and shape plus frequency change experiments (**c**). **(d)** The averages of the distributions in **a**–**c**. For the spatial frequency condition (High to low), the average latency was reduced relative to control, likely due to the few saccades with very express latencies. With shape plus frequency changes, it was increased again, but there was (overall) no significant effect of condition on catch-up saccade latency. A Kruskal-Wallis nonparametric ANOVA across all conditions showed no significance ($p=0.2617$). **(e–h)** Same as **a**–**d** but when landing on a high spatial frequency texture. Here, the catch-up saccade latencies were generally faster than when landing on a low spatial frequency (compare **h** to **d**). Perhaps as a result, it was easier to see the influence of intra-saccadic stimulus changes. Specifically, catch-up saccade latencies increased relative to control (High to high), and there was a significant effect across conditions in **h** ($p=0.0049$). Thus, elevation of foveal SC activity on intra-saccadic change trials was associated with slightly delayed corrective saccades. This is an expected consequence of prediction error signaling and also of elevated foveal SC activity. Error bars denote SEM. Note that the histograms show fewer numbers of catch-up saccades on intra-saccadic change trials when compared to control trials. This is because there were fewer intra-saccadic change trials than control trials in our experiments (Methods). The figure's underlying data are included in S16 Data (the summary statistics in **d**, **h** are those obtained from the raw distributions in the other panels).
(PDF)

**S1 Data. Data underlying Fig 1.**
(XLSX)

**S2 Data. Data underlying Fig 2.**
(XLSX)

**S3 Data. Data underlying Fig 3.**
(XLSX)

**S4 Data. Data underlying Fig 4.**
(XLSX)

**S5 Data. Data underlying Fig 5.**
(XLSX)

**S6 Data. Data underlying Fig 6.**
(XLSX)

**S7 Data. Data underlying Fig 7.**
(XLSX)

**S8 Data. Data underlying Fig 8.**
(XLSX)

**S9 Data. Data underlying Fig 9.**
(XLSX)

**S10 Data. Data underlying S1 Fig.**
(XLSX)

**S11 Data. Data underlying S2 Fig.**
(XLSX)

**S12 Data. Data underlying S3 Fig.**
(XLSX)

**S13 Data. Data underlying S4 Fig.**
(XLSX)

**S14 Data. Data underlying S5 Fig.**
(XLSX)

**S15 Data. Data underlying S6 Fig.**
(XLSX)

**S16 Data. Data underlying S7 Fig.**
(XLSX)

## Author contributions

**Conceptualization:** Tong Zhang, Amarender R. Bogadhi, Ziad M. Hafed.

**Data curation:** Tong Zhang, Amarender R. Bogadhi, Ziad M. Hafed.

**Formal analysis:** Tong Zhang, Ziad M. Hafed.

**Funding acquisition:** Amarender R. Bogadhi, Ziad M. Hafed.

**Investigation:** Ziad M. Hafed.

**Methodology:** Ziad M. Hafed.

**Project administration:** Ziad M. Hafed.

**Supervision:** Ziad M. Hafed.

**Validation:** Ziad M. Hafed.

**Visualization:** Ziad M. Hafed.

**Writing – original draft:** Tong Zhang, Amarender R. Bogadhi, Ziad M. Hafed.

**Writing – review & editing:** Tong Zhang, Amarender R. Bogadhi, Ziad M. Hafed.

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
