## [Editor Report · Decision Letter 0]

Dear Ziad, 

Thank you for submitting your manuscript entitled "Signaling of trans-saccadic prediction error by foveal neurons of the monkey superior colliculus" for consideration as a Research Article by PLOS Biology.

Your manuscript has now been evaluated by the PLOS Biology editorial staff as well as by an academic editor with relevant expertise and I am writing to let you know that we would like to send your submission out for external peer review.

Once your full submission is complete, your paper will undergo a series of checks in preparation for peer review. After your manuscript has passed the checks it will be sent out for review. To provide the metadata for your submission, please Login to Editorial Manager (https://www.editorialmanager.com/pbiology) within two working days, i.e. by Mar 14 2025 11:59PM.

Kind regards,

Christian

Christian Schnell, PhD

Senior Editor

PLOS Biology

cschnell@plos.org

---

## [Decision Letter · Decision Letter 1]

Dear Ziad,

Thank you for your patience while your manuscript "Signaling of trans-saccadic prediction error by foveal neurons of the monkey superior colliculus" went through peer-review at PLOS Biology. Your manuscript has now been evaluated by the PLOS Biology editors, an Academic Editor with relevant expertise, and by several independent reviewers.

In light of the reviews, which you will find at the end of this email, we are pleased to offer you the opportunity to address the comments from the reviewers in a revision that we anticipate should not take you very long. We will then assess your revised manuscript and your response to the reviewers' comments with our Academic Editor aiming to avoid further rounds of peer-review, although we might need to consult with the reviewers, depending on the nature of the revisions.

**IMPORTANT - SUBMITTING YOUR REVISION**

*Resubmission Checklist*

*Published Peer Review*

*PLOS Data Policy*

*Blot and Gel Data Policy*

Sincerely,

Christian

Christian Schnell, PhD

Senior Editor

PLOS Biology

cschnell@plos.org

REVIEWS:

Reviewer #1: The study of Zhang et al. uncovers an interesting line of evidence supporting the role of primate superior colliculus in detecting a change in the visual field during a saccadic eye movement. By using a simple paradigm in which a static visual image undergoes a change contingent upon a saccade start, they show that the neurons in primate SC can encode that change regardless of these visual neurons' own tuning preference for stimulus features. They also demonstrate that the magnitude of this change detection in SC neurons scales with the amount of change in the visual stimulus in shape and spatial frequency. Overall, they report a general trend in the peri-saccadic activity of SC neurons consistent with a mechanism of detecting prediction error. 

The paper overall is excellent. The questions are well-defined, the results are clear, and the statistical methods are sound. The paper is also clearly written. This study addresses an important problem in peri-saccadic visual perception - how does the visual system maintain stability and detect change across saccades? This question has somewhat gone out of fashion in recent neuroscience work because of some controversial findings in the literature. Zhang et al's study offers evidence for a refreshing alternative to approaching the problem. Therefore, this study is a significant contribution to the field. 

I don't have any major issues with the claims and results. If the authors could address the following minor comments, I believe that would improve the paper. 

1. A. The introduction gives the example of the Mona Lisa to motivate the visual stability problem. However, I found the example a bit confusing. My confusion was that when I look at Mona Lisa's eyes and saccade to her mouth, I don't perceive a stable portrait. I perceive that her smile suddenly vanishes. This 'fleeting smile' is what made this artwork so popular. Margaret Livingstone has published a paper explaining this phenomenon (https://doi.org/10.1126/science.290.5495.1299b). The visual system detects a change in the portrait, which is not actually there in real-time. It is simply a function of how visual receptive fields are arranged topographically - low spatial frequency-tuned neurons tile more of peripheral vision than foveal. Maybe the authors can use a different example that doesn't cause this particular confusion. 

B. Conceptually, the framing of the authors' main question needs to explicate certain nuances. In line 638, they say - "our focus here: how is visual sensation of a stably present visual object in the environment handled when an eye movement brings this object inside a given neuron's RF?". Due to the gradient of spatial frequency over a visual retinotopic map, a peripheral object, otherwise stable, will necessarily look different to visual neurons at the fovea. How does the visual system deal with the mismatch of spatial frequency of its filters? I understand that the authors are not addressing this question in their study, but this nuance could be explicitly mentioned to make the discussion richer and avoid confusion. 

2. Along the same lines, maybe the authors can discuss if the visual system uses a similar mechanism to detect a peri-saccadic perceptual change not because of a physical change in the visual field but because of the spatial frequency gradient on the visual images - exactly like the 'fleeting smile' on Mona Lisa. Given the recent findings of short latency responses to faces in primate SC (https://doi.org/10.1016/j.neuron.2024.06.005), I think it is a great opportunity for the authors to link their findings of prediction error in SC to short latency face detection in primate SC. This could be one way to use the same example but more powerfully and less confusingly. But I leave it up to the authors to decide. 

3. The methodology would be more robust if the authors could provide, if possible, a photodiode readout of the frame luminance across time. This is to be absolutely sure that the saccade-triggered change in visual stimulus occurred mid-flight. If the current calculations (I assume based on software and hardware TTLs) could be corroborated with actual events on the screen, all doubts about instrumentation delays would be cleared. 

Reviewer #2: Summary 

The authors identify a potential mismatch signal between expected and experienced spatial frequencies across saccades in foveal neurons of the macaque superior colliculus. In particular, reafferent visual firing rates of foveal SC neurons are slightly higher when the stimulus that is foveated has its spatial frequency changed unexpectedly during a saccade. They find that this phenomenon is unique to the condition where the animal initiates a saccade, and scales with the number of features that are mismatched pre- and post-saccadically (i.e., firing rates for shape + spatial frequency changes are higher than for spatial frequency alone). 

Comments

Since one main goal of making saccades is to foveate targets of interest, this work showing that unexpected events across saccades are processed at the center of gaze is a valuable addition to the literature on the neural basis of visual stability across saccades. 

1) My biggest concern is that the main mismatch effects shown across the population of neurons in Figs. 3, 6, and 7 are small, and I'd want to be sure that they are robust across potential analysis choices. For example, for all the population visual response comparisons, the authors calculate reafferent firing rates aligned to the change event in all conditions including those with a saccade. However, this period includes the offset of the saccade when the eyes are still in flight and a rebound in the neuronal activity after a period of saccadic suppression, both of which may introduce spurious transients in the visual responses. Are the same effects observed even when the reafferent period is aligned to the offset of the saccade rather than the visual change? 

2) The main analyses show the mean firing rates and their modulation across conditions for individual neurons and a statistical comparison of those means across the population. It would also be informative to see what proportion of individual neurons show a significant difference between the change and control conditions across trials. 

3) Were there any systematic differences in eye position or velocity, either as post-saccadic fixational eye movements/drift or during the instructed saccade, when the visual feature changed intrasaccadically? 

4) Finally, in the Discussion section, the authors speculate on ways in which foveal neurons may have access to the pre-saccadic visual information in order to identify the difference between pre- and post-saccadic stimuli. These include neurons with larger receptive fields and visual information carried by motor neurons. The most straightforward possibility in my opinion is that of pre-saccadic remapping, which the authors mention but reject as a possible explanation. I did not follow the reasoning for why this is a conceptually unrelated phenomenon. If a visually responsive cell could, via remapping, briefly sample the part of the visual field that it will occupy after a saccade, would that not give it direct access to both the pre- and post-saccadic visual stimulus? However, this is only a relatively minor point of interpretation that does not change the presentation of the main observations here. 

Reviewer #3: How our visual system keeps a stable representation of the visual scene during saccadic eye movements is a crucial piece of the puzzle of how in a dynamically changing world our brain constructs a stable mental model resilient to changes in the flow of visual signals and other interruptions. In this study Zhang, Bogadhi and Hafed study how fovea representing neurons within the superior colliculus (SC) are modulated by information presented peripherally, when that information is going to be brought up to the fovea during an upcoming eye movement. It should be noted that, in spite of the significance of the fovea, in general in visual system electrophysiology experiments, the foveal representation is historically understudied and this also enhances the significance of the current paper. The manuscript is very well-written and the research questions are significant and properly studied. I did not find any major issues with the experimental design, analysis or interpretation. Thus, there are only a few minor comments as listed below. However, I have a few comments, more reflecting of missed opportunities rather than necessary analysis, and I totally leave it to the authors to decide if they want to address them in this paper or leave it to be addressed in future manuscripts. 

Comment1: As pointed out by the authors, the transfer of information across various representations has been reported in several contexts. In spite the rich literature, there are not many studies investigating the neural mechanisms involved in such "rerouting of information". It would be great to know the authors' perspective on the neural machinery involved in such rerouting of information in the discussion section. Also, if their dataset allows, investigating how propagation of oscillatory activity might be coupled with information rerouting (e.g. similar to Neupane, Guitton, Pack PNAS 2017) could be a very good extension of the current study. Again, leaving it for the authors to decide when and how to dig deeper into the neural mechanisms involved.

Comment 2) Coding multiple locations can also be interpreted as having "mixed selectivity". Population-level analyses such as PCA have been shown to reveal hidden aspects of the neural code where single neuron analysis is missing to trace them. Even non-simultaneously recorded neurons can be used to trace how much information at the population level does exist regarding a peripheral target and how this information changes around the time of saccade. Again, I would leave it to the authors to decide whether they want to employ population-level analysis for this manuscript or not. 

Comment 3) I was hopeful to see a figure depicting the tiling of the receptive fields around the fovea, or at least their centers. 

Comment 4) It seems imperative to show the normalized response of population of neurons (equivalent of figure 2b, c but for the population) before jumping to scatter plot-level description in figure 3. In general, the temporal dynamics at the population level is missing for almost all reported phenomena and it would increase the value of the paper to bridge the single neuron temporal response and population-level histograms/scatter plots with a population-level normalized response; and even preferably dynamics of an index such as ROC or modulation index across time. 

Minor comments: 

- Isn't it better to call it e.g. psuedosaccade in fig. 5? Calling it saccade might be confusing. 

- Putting p values in the figures right above where the sample sizes are reported (e.g. n=59) could help the reader grasp the significance of the figures quicker.

---

## [Editor Report · Decision Letter 2]

Dear Ziad,

Thank you for your patience while we considered your revised manuscript "Signaling of trans-saccadic prediction error by foveal neurons of the monkey superior colliculus" for publication as a Research Article at PLOS Biology. This revised version of your manuscript has been evaluated by the PLOS Biology editors and the Academic Editor.

Based on our Academic Editor's assessment of your revision, we are likely to accept this manuscript for publication, provided you satisfactorily address the following data and other policy-related requests:

* We would like to suggest a different title to improve its accessibility for our broad audience: 

Foveal neurons of the monkey superior colliculus signal trans-saccadic prediction errors

* Please add the links to the funding agencies in the Financial Disclosure statement in the manuscript details.

* DATA POLICY:

Regardless of the method selected, please ensure that you provide the individual numerical values that underlie the summary data displayed in the following figure panels as they are essential for readers to assess your analysis and to reproduce it: S7DH

* CODE POLICY

We expect to receive your revised manuscript within two weeks. 

*Published Peer Review History*

*Press*

Sincerely,

Christian

Christian Schnell, PhD

Senior Editor

cschnell@plos.org

PLOS Biology

---

## [Editor Report · Decision Letter 3]

Dear Ziad,

Thank you for the submission of your revised Research Article "Foveal neurons of the monkey superior colliculus signal trans-saccadic prediction errors" for publication in PLOS Biology. On behalf of my colleagues and the Academic Editor, Christopher Pack, I am pleased to say that we can in principle accept your manuscript for publication, provided you address any remaining formatting and reporting issues. These will be detailed in an email you should receive within 2-3 business days from our colleagues in the journal operations team; no action is required from you until then. Please note that we will not be able to formally accept your manuscript and schedule it for publication until you have completed any requested changes.

PRESS

Sincerely, 

Christian

Christian Schnell, PhD

Senior Editor

PLOS Biology

cschnell@plos.org